# An open source and reduce expenditure ROS generation strategy for chemodynamic/ photodynamic synergistic therapy

Conghui Liu [1,2], Yu Cao[1], Yaru Cheng[1], Dongdong Wang[1], Tailin Xu [1], Lei Su [1], Xueji Zhang[1,2✉] & Haifeng Dong [1✉]

The therapeutic effect of reactive oxygen species (ROS)-involved cancer therapies is significantly limited by shortage of oxy-substrates, such as hypoxia in photodynamic therapy (PDT) and insufficient hydrogen peroxide ($H_2O_2$) in chemodynamic therapy (CDT). Here, we report a $H_2O_2/O_2$ self-supplying nanoagent, (MSNs@$CaO_2$-ICG)@LA, which consists of manganese silicate (MSN)-supported calcium peroxide ($CaO_2$) and indocyanine green (ICG) with further surface modification of phase-change material lauric acid (LA). Under laser irradiation, ICG simultaneously generates singlet oxygen and emits heat to melt the LA. The exposed $CaO_2$ reacts with water to produce $O_2$ and $H_2O_2$ for hypoxia-relieved ICG-mediated PDT and $H_2O_2$-supplying MSN-based CDT, acting as an open source strategy for ROS production. Additionally, the MSNs-induced glutathione depletion protects ROS from scavenging, termed reduce expenditure. This open source and reduce expenditure strategy is effective in inhibiting tumor growth both in vitro and in vivo, and significantly improves ROS generation efficiency from multi-level for ROS-involved cancer therapies.

[1] Beijing Advanced Innovation Center for Materials Genome Engineering, University of Science & Technology Beijing, Beijing 100083, P. R. China. [2] School of Biomedical Engineering, Health Science Center, Shenzhen University, Shenzhen, Guangdong 518060, P. R. China. ✉email: zhangxueji@ustb.edu.cn; hfdong@ustb.edu.cn

Reactive oxygen species (ROS), mainly including active superoxide anions ($O_2^-$), hydroxyl radicals ($\bullet OH$), and singlet oxygen ($^1O_2$), act as significant signaling and regulatory molecules at physiologic levels, conversely, will damage cells once the concentration elevate at an abnormal level[1]. Elevated ROS level is one of the characteristics of tumor micro-environment (TME), and along with high ROS elimination rates exist in cancer cells to maintain a steady equilibrium state, called self-adaptation mechanisms. Thus, cancer cells are more sensitive to further enhanced oxidative stress beyond the cellular tolerability threshold[2]. On this basis, ROS-mediated therapies, such as photodynamic therapy (PDT)[3–6] and chemodynamic therapy (CDT)[7–11], are developed to disrupt the cellular self-adaptation mechanisms and induce cell death based on ROS-generating agents[12].

The PDT utilizes light-activated photosensitizers to convert oxygen ($O_2$) to ROS[13], whereas CDT takes advantage of an in situ Fenton or Fenton-like reaction between hydrogen peroxide ($H_2O_2$) and catalysts to generate cytotoxic hydroxyl radical ($\bullet OH$)[14,15]. Recently, the PDT/CDT combination therapy has been continuously explored to amplify the tumor oxidative stress and achieve better anticancer therapeutic effect than monotherapy[16–20]. However, the TME feature of hypoxia, depletable amount of $H_2O_2$ and the glutathione (GSH) depletion effect on ROS still limit ROS efficiency[15,21,22]. Two different feasible strategies have been proposed to relieve hypoxia in PDT and supplement the cellular amount of $H_2O_2$ in CDT, respectively, amplifying endogenous $O_2$/$H_2O_2$ generation[11,23–26] or directly delivering exogenous $O_2$/$H_2O_2$ into cells[27–29]. To date, there have indeed been some nanosystems for synergistic PDT/CDT, but most of them only overcome part of the limitations. For example, Liu et. al constructed sorafenib@$Fe^{3+}$-tannic acid nanoparticles with GSH depletion property for PDT/CDT[19]. Copper ferrite nanospheres[18], copper/manganese silicate nanospheres[16], and ROS-activatable liposomes[30] have been reported for hypoxia-relieved and GSH-depleting synergistic PDT/CDT. At present, simultaneous hypoxia relief, $H_2O_2$ supplement, and GSH-depletion nanosystems have been little reported, which is highly desirable in PDT/CDT combination therapy. $CaO_2$, a safe solid inorganic peroxide, can decompose to simultaneously release $O_2$ and $H_2O_2$ in contacting with water[31] and have been widely applied in remediation of environmental contamination[32]. Therefore, the introduction of $CaO_2$ into ROS-involved therapies seem to hold great promise for enhanced ROS generation.

Herein, a $H_2O_2$/$O_2$ self-supplying thermoresponsive nanosystem, (MSNs@$CaO_2$-ICG)@LA, consisting of manganese silicate (MSNs) supported calcium peroxide ($CaO_2$) nanoparticles (NPs) and indocyanine green (ICG) with further surface coating of a phase-change material lauric acid (LA, melting point: 44~46 °C), is reported for photodynamic/chemodynamic synergistic cancer therapy (Fig. 1). In this nanosystem, the $CaO_2$ is protected from water by LA until the outer layer LA is melted owing to the photothermal effect of ICG under the irradiation of a near-infrared (NIR) 808 nm laser. The exposed $CaO_2$ reacts with water to rapidly generate $H_2O_2$ and $O_2$, and accompanies exposure of inner MSNs. The released $O_2$ can relieve hypoxia for enhanced ICG-mediated PDT. The interaction between MSNs and GSH lead to release of Fenton-like agent $Mn^{2+}$ for $H_2O_2$-supplementing CDT and magnetic resonance imaging (MRI)-guided therapy. This GSH depletion further enhance the ROS generation efficiency. Thus we report a smart system (MSNs@$CaO_2$-ICG)

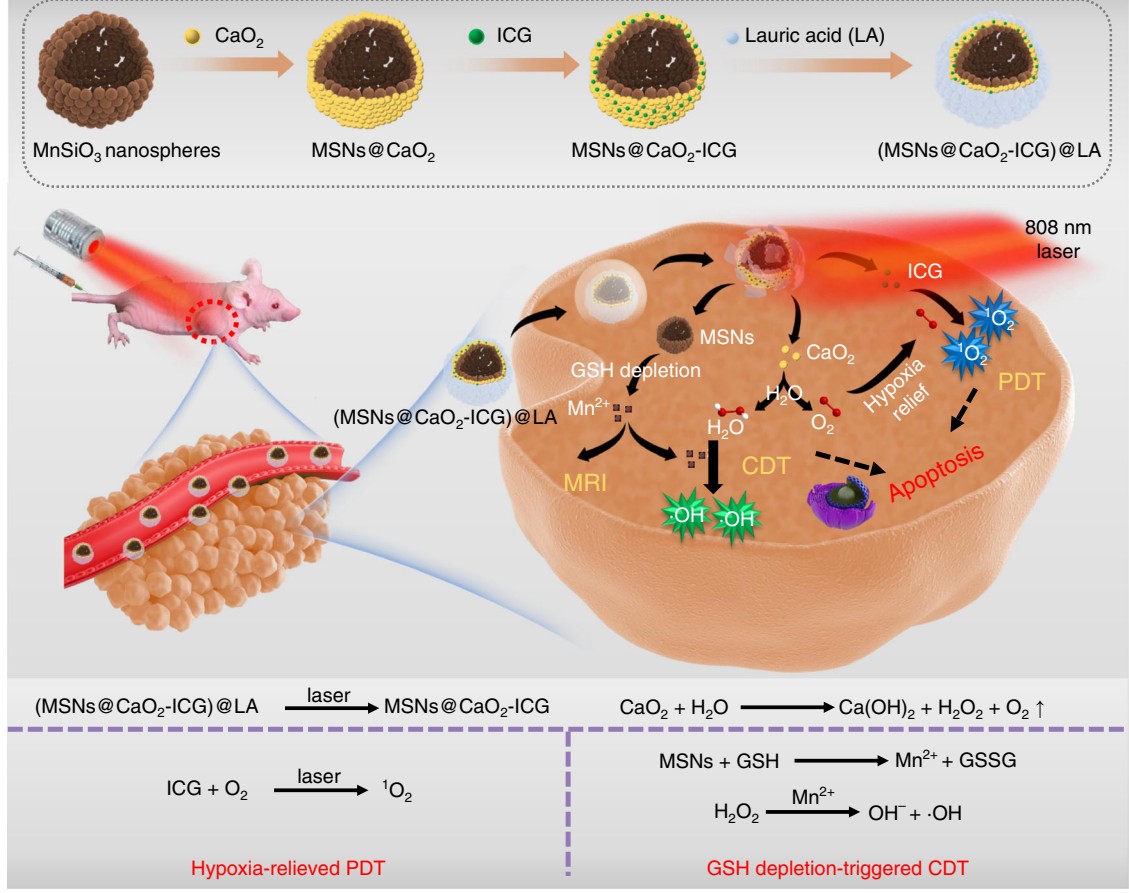

$$(MSNs@CaO_2\text{-}ICG)@LA \xrightarrow{\text{laser}} MSNs@CaO_2\text{-}ICG$$

$$CaO_2 + H_2O \longrightarrow Ca(OH)_2 + H_2O_2 + O_2 \uparrow$$

$$ICG + O_2 \xrightarrow{\text{laser}} {}^1O_2$$

$$MSNs + GSH \longrightarrow Mn^{2+} + GSSG$$

$$H_2O_2 \xrightarrow{Mn^{2+}} OH^- + \cdot OH$$

Hypoxia-relieved PDT          GSH depletion-triggered CDT

**Fig. 1 Open source and reduce expenditure ROS generation strategy.** The scheme of fabrication process and therapeutic mechanism of thermo-responsive (MSNs@$CaO_2$-ICG)@LA NPs for synergistic CDT/PDT with $H_2O_2$/$O_2$ self-supply and GSH depletion.

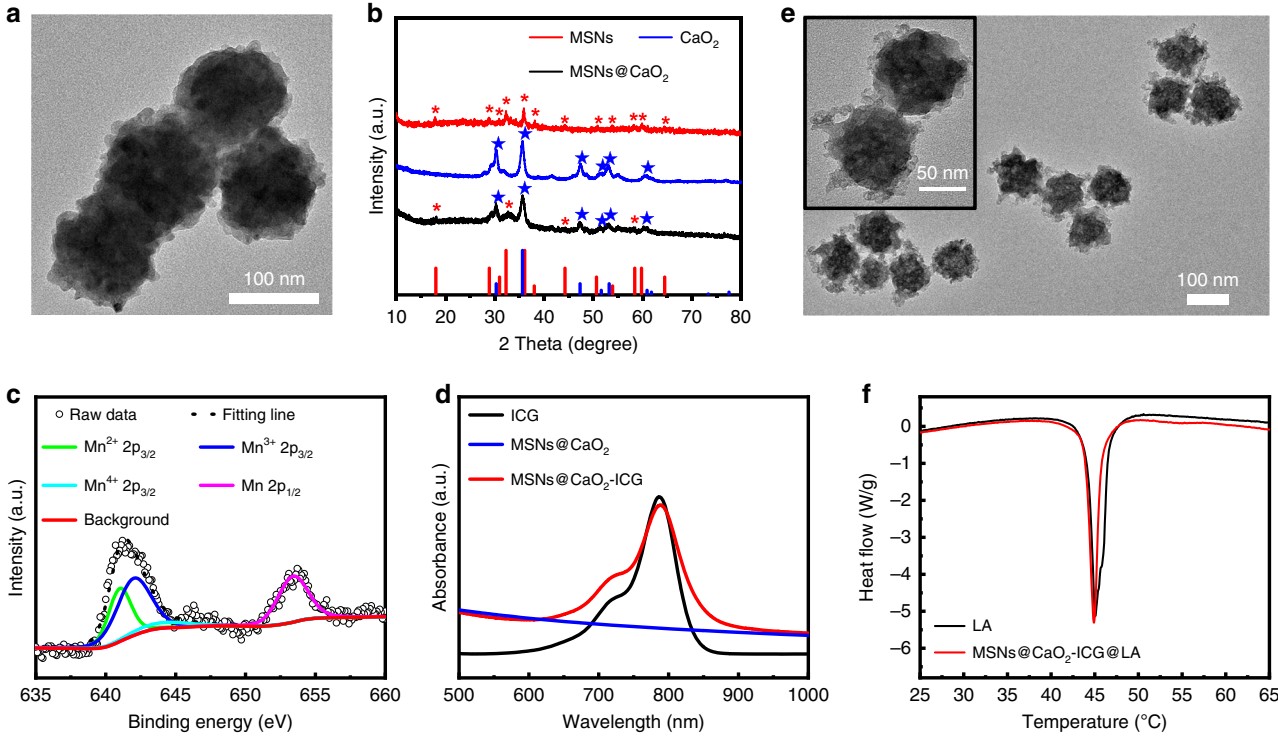

**Fig. 2 Physiochemical characterizations. a** TEM images of MSNs@CaO₂ NPs. **b** XRD spectra of MSNs, CaO₂ NPs, and MSNs@CaO₂. **c** High-resolution Mn 2p XPS spectra of MSNs@CaO₂. **d** UV-vis spectra of ICG, MSNs@CaO₂, and MSNs@CaO₂-ICG. **e** TEM images of (MSNs@CaO₂-ICG)@LA. **f** DSC curves of LA and (MSNs@CaO₂-ICG)@LA.

@LA can simultaneously overcome the main limitations including hypoxia, depletable amount of $H_2O_2$ and GSH elimination effect on ROS for synergistic PDT/CDT, and this open source and reduce expenditure ROS-produced way obtain excellent tumor inhibition effect both in vitro and in vivo, provide a universal idea of therapeutic nanoagents design for synergistic PDT/CDT.

## Results

**Preparation and characterization of (MSNs@CaO₂-ICG)@LA.** The MSNs with an average diameter of 120 nm were synthesized by a hydrothermal method (Supplementary Figure 1a). The CaO₂ NPs with an average diameter of 20 nm were prepared through a hydrolysis-precipitation process (Supplementary Figure 1b)[33,34]. Then, the CaO₂ NPs were assembled onto the surface of MSNs to form MSNs@CaO₂ NPs in methanol through electrostatic adsorption, confirmed by the transmission electron microscope (TEM) images (Fig. 2a) and X-ray diffraction (XRD) pattern with the characteristic peaks belong to MSNs and CaO₂, respectively (Fig. 2b). The successful assembly of MSNs@CaO₂ was also validated by the X-ray photo-electron spectroscopy (XPS) analysis (Supplementary Figure 2a), and the high resolution XPS revealed that the O 2 s existed primarily in the form of silicate and peroxo groups (Supplementary Figure 2b), and the Mn $2p_{3/2}$ mainly consisted of 34.65% $Mn^{2+}$ (641 eV), 55.74% $Mn^{3+}$ (642 eV), and 9.61% $Mn^{4+}$ (644 eV) (Fig. 2c)[11,16]. The high content of $Mn^{3+}$ in MSNs made it possible to react with endogenous GSH for further biodegradation[16,35], and the TEM analysis confirmed the gradual biodegradation of MSNs in the presence of GSH (Supplementary Figure 3). Thus, MSNs could degrade to release $Mn^{2+}$ for CDT by depleting ROS scavenger GSH, which was beneficial to amplify the therapeutic effect[36].

ICG is a NIR tricarbocyanine dye approved by the US Food and Drug Administration for clinical use and show great potential both in PDT and PTT[37,38]. We further incorporated ICG into the nanosystem to utilize its photothermal property and $^1O_2$ generation capacity. The strong absorption peak centered at 800 in UV–vis spectra of MSNs@CaO₂-ICG validated the integration of ICG molecules into the nanosystem, and the loading content of ICG was estimated to be 7.87 wt % (Fig. 2d). The phase-change material LA with good biocompatibility and biodegradability[39,40] was added as coating to obtain (MSNs@CaO₂-ICG)@LA with thermo-responsive property. The TEM image of (MSNs@CaO₂-ICG)@LA showed the uniform size (Fig. 2e) and the change of surface zeta potential (Supplementary Figure 4) confirmed the preparation process. The differential scanning calorimetry (DSC) curves of (MSNs@CaO₂-ICG)@LA exhibited a similar melting point to pure LA, validating the successful coating of LA on MSNs@CaO₂-ICG (Fig. 2f). The resulting (MSNs@CaO₂-ICG)@LA also showed good stability in water, PBS (10 mM, pH 7.4) and Dulbecco's modified eagle medium (DMEM) solution demonstrated by the hydrodynamic particle size and surface zeta potential analysis after 24 h (Supplementary Figure 5).

**In vitro $H_2O_2$ and $O_2$ generation and thermo-responsive property.** Cumulative amount of $H_2O_2$ released from CaO₂ was measured by Cu(II)-neocuproine spectrophotometric method[41]. Pure CaO₂ immediately reacted with water to generate $H_2O_2$ up to 75 μM at 25 °C, whereas the cumulative $H_2O_2$ amount of CaO₂@LA remained at the same level lower than 30 μM within 7 h at 25 °C (Fig. 3a), suggesting the LA coating can protect CaO₂ from reacting with water beforehand. In addition, the $H_2O_2$ releasing was increased when the pH descended, suggesting the acidic environment in tumors was good for the $H_2O_2$ generation (Supplementary Figure 6). When the temperature increase to 46 °C, the CaO₂@LA recovered the rapid $H_2O_2$ generation ability similar to pure CaO₂ owing to the melting of LA coating. The CaO₂@LA solution at 25 °C displayed no significant change in dissolved $O_2$ level within 15 min monitored by a portable

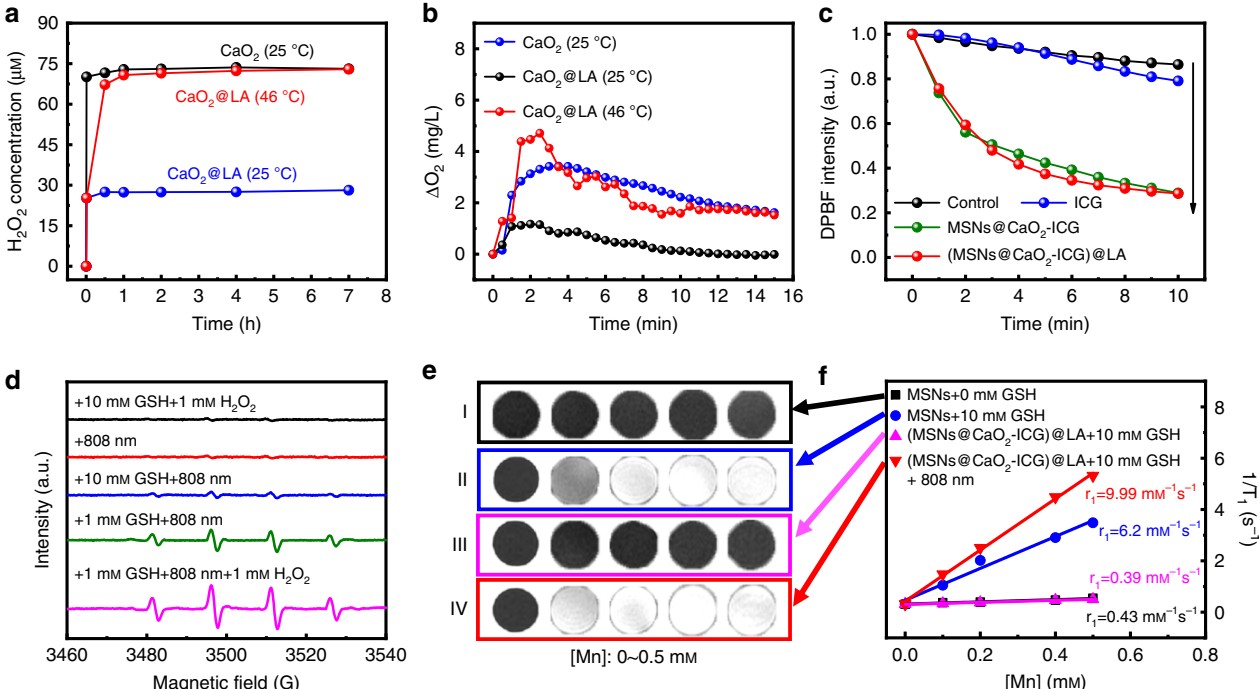

**Fig. 3 ROS generation and in vitro MRI. a** $H_2O_2$ cumulative release profile ([$CaO_2$]=10 μg mL$^{-1}$) and **b** $O_2$ concentration measurement in PBS (10 mM, pH = 7.4) ([$CaO_2$]=100 μg mL$^{-1}$). **c** Time-dependent degradation of DPBF irradiated by laser for 10 min. ([MSNs] = 50 μg mL$^{-1}$, [ICG] = 8 μg mL$^{-1}$. Laser: 808 nm, 0.64 W cm$^{-2}$). **d** EPR analysis of •OH production in each group with different treatment. DMPO was used as the spin-trapping agent. The sample in each group was (MSNs@$CaO_2$-ICG)@LA aqueous solution ([MSNs] = 50 μg mL$^{-1}$). **e** In vitro MRI of different solution and **f** the corresponding $r_1$ value ([MSNs] = 50 μg mL$^{-1}$. Laser: 808 nm, 0.64 W cm$^{-2}$, 10 min).

dissolved oxygen meter in real time (Fig. 3b). However, when heated to 46 °C, the $O_2$ concentration showed a rapid increase in the first 3 min faster than pure $CaO_2$ at 25 °C, then fell slowly. These results fully demonstrated the $H_2O_2$ and $O_2$ simultaneous generation capacities of $CaO_2$ NPs in the reaction with water, and preliminarily verified the thermalresponsive property of LA.

**Enhanced ROS generation measurements of (MSNs@$CaO_2$-ICG)@LA.** The (MSNs@$CaO_2$-ICG)@LA exhibited an ICG concentration-dependent photothermal effect under an 808 nm irradiation (Supplementary Figure 7). Upon irradiation, the temperature increase induced melting of LA (Supplementary Figure 8), and the ICG was gradually released from (MSNs@$CaO_2$-ICG)@LA (Supplementary Figure 9), demonstrating the successful design of this thermoresponsive nanosystem. The $^1O_2$ generation of (MSNs@$CaO_2$-ICG)@LA in vitro was monitored using 1,3-diphenylisobenzofuran (DPBF) as chemical probe. The DPBF content of ICG, MSNs or $CaO_2$ group displayed slight decline compared with the control group under the irradiation of 808 nm laser (Fig. 3c and Supplementary Figure 10a). The temperature rise and NIR laser irradiation alone showed negligible effect on the degradation of DPBF (Supplementary Figure 10b). In contrast, the DPBF treated with MSNs@$CaO_2$-ICG and (MSNs@$CaO_2$-ICG)@LA showed a sharp decrease within 10 min owing to $O_2$ self-supplying PDT effect of ICG. The •OH generation ability was evaluated by methylene blue (MB) degradation. The redox reaction between MSNs and GSH induced $Mn^{2+}$ release and GSH depletion, and the $Mn^{2+}$ reacted with $H_2O_2$ to produce active •OH through Fenton-like reaction (Supplementary Figure 11a). It was worthy to mention that the MB degradation increased with the increase of GSH concentration from 0 to 1.0 mM, but decreased when further increased GSH concentration as excessive GSH would scavenge •OH conversely (Supplementary Figure 11b). The $H_2O_2$ concentration-dependent Fenton-like effect provided the great possibility of enhanced •OH generation by $H_2O_2$

self-supply from $CaO_2$ (Supplementary Figure 11c). Similar results to the MB degradation experiments were also obtained by electron paramagnetic resonance (EPR) analysis of •OH production of (MSNs@$CaO_2$-ICG)@LA as shown in Fig. 3d.

The released $Mn^{2+}$ from MSNs could also be utilized as MRI contrast agent. As shown in Fig. 3e, the $T_1$ signal intensity of MSNs (group I) had negligible change and the released $Mn^{2+}$ were at very low concentration, whereas MSNs treated with 10 mM GSH (group II) exhibited enhanced brightness derived from paramagnetic $Mn^{2+}$ centers because MSN was reduced by GSH and the increasing free Mn ions (Supplementary Table 1) were easier to proceed chemical exchange with protons than isolated Mn centers in MSNs for enhanced $T_1$ signal[42]. Thus, it was rational that (MSNs@$CaO_2$-ICG)@LA displayed enhanced brightness only when co-treated with GSH and NIR laser irradiation (group IV). Remarkably, the longitudinal relaxivity coefficient ($r_1$) of group IV and correspongding released Mn concentration were larger than that of group II (Fig. 3f). This was attributed to the enhanced release of $Mn^{2+}$ from MSNs accelerated by photothermal effect of ICG (Supplementary Table 1).

**Intracellular uptake of (MSNs@$CaO_2$-ICG)@LA.** Before evaluating the feasibility of (MSNs@$CaO_2$-ICG)@LA for in vivo antitumor therapy, the cytotoxicity and cell uptake of (MSNs@$CaO_2$-ICG)@LA were first investigated. As shown in Fig. 4a, the (MSNs@$CaO_2$-ICG)@LA showed little cytotoxicity toward MCF-7, A549, and NHDF cells when the concentration was from 0 to 50 μg mL$^{-1}$ after incubation for 12 h, indicating good biocompatibility. The flow cytometry was conducted to measure the fluorescence intensity of ICG in MCF-7 cells treated with (MSNs@$CaO_2$-ICG)@LA at different incubation time point (Fig. 4b), and the corresponding analysis of mean fluorescence intensity was shown in Fig. 4c. Comparing with the blank control

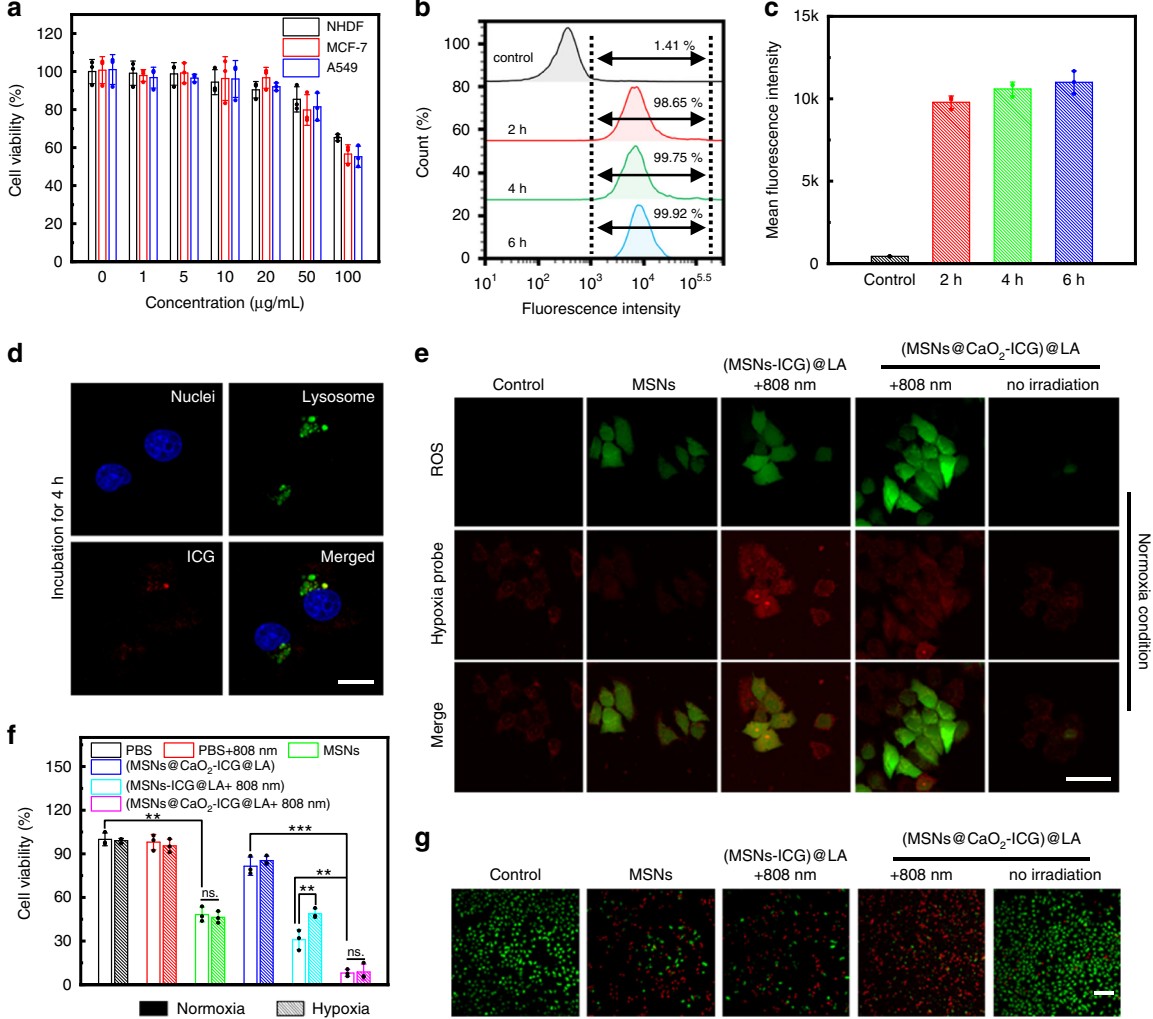

**Fig. 4 Intracellular uptake and ROS generation of (MSNs@CaO₂-ICG)@LA. a** Cell viability of MCF-7, A549, and NHDF cells treated with different concentrations of (MSNs@CaO₂-ICG)@LA by CCK-8 assay. Data are presented as mean ± SD ($n = 3$). **b** The rate of MCF-7 cells uptaking (MSNs@CaO₂-ICG)@LA ([MSNs] = 25 μg mL⁻¹), and **c** corresponding mean fluorescence intensity by flow cytometry after different incubation time. Data are presented as mean ± SD ($n = 3$). **d** Fluorescence images showing cellular uptake of (MSNs@CaO₂-ICG)@LA ([MSNs] = 25 μg mL⁻¹) in MCF-7 cells after incubation for 4 h. Scale bar: 20 μm. **e** Fluorescence images showing ROS and hypoxia level in MCF-7 cells with different treatment under normoxia condition. ([MSNs] = 25 μg mL⁻¹. Laser: 0.64 W cm⁻², 10 min). Scale bar: 50 μm. **f** Cell viability of MCF-7 cells with different treatments under hypoxia or normoxia condition ([MSNs] = 25 μg mL⁻¹. Laser: 0.64 W cm⁻², 10 min). The mean value was calculated by the two-tailed $t$ test (mean ± SD, $n = 3$). **P < 0.01 and ***P < 0.001, compared with the indicated group. **g** Fluorescence images of Calcein-AM- and propidium iodide (PI)-costained MCF-7 cells with different treatments under normoxia condition ([MSNs] = 25 μg mL⁻¹. Laser: 0.64 W cm⁻², 10 min). Scale bar: 200 μm.

group, the cells treated with (MSNs@CaO₂-ICG)@LA demonstrated high uptake rate as the incubation time extended, and incubation for 4 h was sufficient. Figure 4d demonstrated that the location of (MSNs@CaO₂-ICG)@LA in MCF-7 cells overlapped with lysosome after incubation for 4 h, suggesting the endolysosomal pathway.

**Intracellular enhanced ROS production**. To investigate the enhanced ROS production of (MSNs@CaO₂-ICG)@LA in living cells, we first explored the intracellular O₂ self-supplying property of CaO₂. As shown in Supplementary Figure 12, after hypoxia treatment, the red fluorescence intensity related to hypoxia of cells nearly remained unchanged when incubated with CaO₂@LA, while that of CaO₂-treated group was significantly weaken, confirming the intracellular hypoxia relief by CaO₂. The CaO₂-mediated O₂ self-supply provided the possibility to enhance the ROS production and PDT efficiency. As expected, the intracellular fluorescence imaging by ROS probe (DCFH-DA, green) and hypoxia probe demonstrated

that the (MSNs@CaO₂-ICG)@LA-treated cells with irradiation displayed the strongest green fluorescence and slight enhanced red fluorescence compared with other groups both under normoxia or hypoxia conditions (Fig. 4e and Supplementary Figure 13), indicating tremendous ROS generation enhanced by O₂ and H₂O₂ self-supply from CaO₂. In order to further investigate the enhanced CDT/PDT therapeutic effect, we used MCF-7 cancer cells as model cell to examined the anticancer effect of (MSNs@CaO₂-ICG)@LA in normoxia as well as hypoxia environment as contrast. Upon a 808 nm laser irradiation, the (MSNs@CaO₂-ICG)@LA treatment showed greatly enhanced cytotoxicity than both MSNs and (MSNs-ICG)@LA in normoxia environment (Fig. 4f), in consistent with the calcein-AM and propidium iodide (PI) co-staining results that most of cancer cells were killed when treated with (MSNs@CaO₂-ICG)@LA plus irradiation (Fig. 4g). Notably, the hypoxia condition had obvious influence on the cell killing ability of (MSNs-ICG)@LA-treated group, whereas that of (MSNs@CaO₂-ICG)@LA-treated

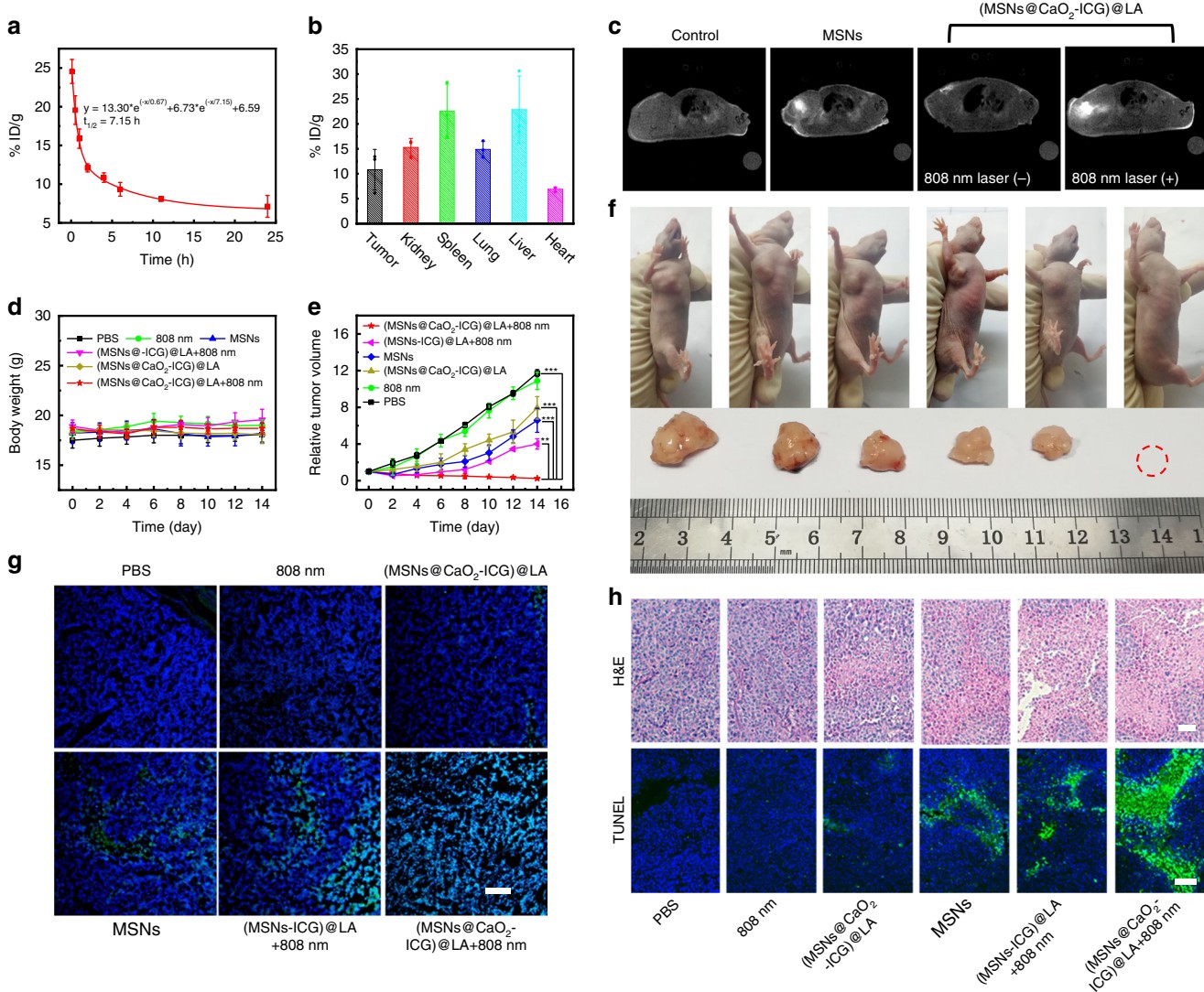

**Fig. 5 In vivo demonstration of synergistic CDT/PDT. a** Blood circulation and **b** biodistribution of (MSNs@CaO$_2$-ICG)@LA by measuring Mn concentrations over a span of 24 h after intravenous injection. Data are presented as mean ± SD ($n = 3$). **c** In vivo MRI of MCF-7 tumor-bearing mice with different treatments after 24 h (Laser: 808 nm, 0.64 W cm$^{-2}$, 10 min). **d** The body weight and **e** the relative cancer volume changes of MCF-7 tumor-bearing mice after various treatments in 14 days. The mean value was calculated by the two-tailed $t$ test (mean ± SD, $n = 5$). **$P < 0.01$ and ***$P < 0.001$, compared with the indicated group. **f** Representative photos of the mice with different treatments and corresponding tumor tissues collected from different groups at 14 days. **g** DCFH (green)- and DAPI (blue)-costained tumor tissues collected from different groups at 24 h post injection. Scale bar: 200 μm. **h** H&E- or TUNEL-stained tumor slices collected from different groups at 14 days. Scale bar: 100 μm.

group almost the same both in normoxia and in hypoxia conditions owing to the O$_2$ self-supplying property of CaO$_2$ (Fig. 4f).

**In vivo tumor treatment efficiency of (MSNs@CaO$_2$-ICG)@LA.**
The antitumor efficacy of (MSNs@CaO$_2$-ICG)@LA-mediated CDT/PDT synergistic therapy in vivo was further conducted in MCF-7 tumor-bearing mice. Figure 5a exhibited the blood circulation curve of (MSNs@CaO$_2$-ICG)@LA in mice by measuring the concentrations of Mn$^{2+}$ in the blood at different time points after tail vein injection, and the fairly long circulation time was favorable for effective tumor accumulation of (MSNs@CaO$_2$-ICG)@LA in tumor site due to the EPR effect which was observed in Fig. 5b. After irradiation by 808 nm laser for 10 min, the tumor-site temperature of (MSNs@CaO$_2$-ICG)@LA-treated mice gradually increased to 46.8 °C, which exceeded the melting point temperature of LA, whereas the control group treated with 808 nm laser-irradiation only did not display obvious

temperature change in tumor site (Supplementary Figure 14). We also investigated the MRI property of (MSNs@CaO$_2$-ICG)@LA in vivo. As shown in Fig. 5c, the brightness in tumor site of mice treated with MSNs was obviously enhanced than the control group, indicating the MSNs could degrade to Mn$^{2+}$ by GSH in vivo. As for (MSNs@CaO$_2$-ICG)@LA-treated mice, the $T_1$ signal intensity became very strong after irradiation of an 808 nm laser, confirming the thermo-responsive property of the nanosystem again. The in vivo MRI property of (MSNs@CaO$_2$-ICG) @LA provided a powerful tool for guiding and monitoring therapy. Then, we divided the MCF-7 tumor-bearing nude mice into six group ($n = 5$) to investigate the antitumor efficacy of different treatments. The tumor volumes of all groups were recorded every 2 days and the tumors were collected at 14 days. As show in Fig. 5d and Supplementary Figure 15, relatively stable body weight and histopathological normal hematoxylin and eosin (H&E) staining of main organs in all groups suggested the negligible side effects of these treatments on mice. The 808 nm laser

irradiation alone displayed little inhibition effect on tumor growth similar to the control group (Fig. 5e, f). The MSNs and showed some tumor repression effects owing to the CDT therapeutic efficacy. The tumors were significantly suppressed in the group of (MSNs-ICG)@LA and an 808 nm laser due to the therapeutic effect of CDT/PDT, and the tumors in the group treated with (MSNs@CaO$_2$-ICG)@LA and an 808 nm laser irradiation were completely eliminated owing to the strongest ROS generation in H$_2$O$_2$/O$_2$ self-supplying CDT/PDT (Fig. 5g and Supplementary Figure 16). The corresponding H&E and TUNEL staining of tumor slides also displayed the maximum tumor necrosis and apoptosis (Fig. 5h). These results indicated that (MSNs@CaO$_2$-ICG)@LA have excellent antitumor effects due to combined PDT/CDT with H$_2$O$_2$ and O$_2$ self-supply.

## Discussion

In summary, we rationally designed a thermo-responsive nanosystem through simple assembly method for overcoming the insufficient supply of O$_2$ and H$_2$O$_2$ in antitumor PDT and CDT. The (MSNs@CaO$_2$-ICG)@LA nanosystem was constructed by decorating CaO$_2$ and ICG on the MSN-support with further surface modification of phase-change material LA. We demonstrated that phase-change material LA could be melted owing to the photothermal effect of excited ICG under irradiation, and then CaO$_2$ could be exposed to react with water to release O$_2$ and H$_2$O$_2$ to enhance the ROS generation in ICG-mediated PDT and Mn$^{2+}$-involved CDT, respectively, acting as an open source strategy for ROS production. The Fenton-like agent Mn$^{2+}$ released out from MSNs depleted ROS scavenger GSH, which further reduced ROS wastage termed reduce expenditure. Our results showed excellent tumor eradication effect of (MSNs@CaO$_2$-ICG)@LA owing to the combined PDT/CDT with O$_2$/H$_2$O$_2$ self-supply and GSH depletion. This work demonstrated a promising open source and reduce expenditure strategy for ROS generation enhancement in ROS-involved cancer therapies.

## Methods

**Instruments**. The morphologies of samples were characterized by TEM (HT7700, Hitachi, Japan). XRD was conducted by a Bruker D8 ADVANCE X. DSC measurements were conducted by DSC Q2000 (TA Instruments, USA) under nitrogen atmosphere. Dissolved oxygen measurement was conducted by JPBJ-608 (Rex, INESA Scientific Instrument). The cell fluorescence imaging was obtained by confocal laser scanning fluorescence microscope (CLSM, FV1200, Olympus, Japan). In vitro and in vivo MRI was performed by an animal MRI scanner (BioSpec70/20USR, Bruker, Germany) at 7.0 T with a gradient echo sequence (TR = 299 ms and TE = 6.01 ms).

**Synthesis of manganese silicate nanoparticles (MSNs)**. The previously reported dendritic mesoporous silica nanoparticles[33] were used as self-killing templates to synthesize MSNs. First, dendritic mesoporous silica nanoparticles (50 mg), MnCl$_2$·4H$_2$O (158.34 mg) and NH$_4$Cl (534.9 mg) were dispersed in 40 mL of water. Then NH$_3$·H$_2$O (28% wt., 1 mL) was added dropwise into the mixture under continuously stirring. After stirring for 30 min, the mixture was transferred into a Teflon-lined autoclave and maintained at 180 °C for 24 h. The precipitates were washed with water and finally dried at 60 °C.

**PEGylation of MSNs**. In brief, MSNs (1 mg) were mixed with methoxy PEG silane (Mw = 2000, 5 mg) in 10 mL of ethanol under magnetic stirring at 60 °C for 24 h. Then, the MSNs-PEG were washed with ethanol and ultrapure water several times.

**Synthesis of (MSNs@CaO$_2$-ICG)@LA**. The CaO$_2$ nanoparticles were synthesized according to previous literature[34]. In brief, ultrapure water (3 mL), calcium chloride (2.7 mmol), ammonia solution (1.5 mL, 1 M) and PEG400 (12 mL) were successively added in the round bottom flask under stirring. Then H$_2$O$_2$ (35 wt. %, 1.5 mL) was dropwise added to the mixture within 50 min. After further stirring for 2 h at 900 rpm at room temperature, the pH of mixture was adjusted to 11.5 using NaOH solution (0.1 M). The precipitate was washed three times with NaOH (0.1 M) and with distilled water until the pH of supernatant reached to 8.4. The precipitate was dried in vacuo at 80 °C for 2 h, then resuspended in ethanol and filtered using a

Millex Filter Unit (0.45 µm). The resulting filtrate was concentrated and dried to achieve CaO$_2$ nanoparticles. MSNs (1 mg) and CaO$_2$ (1 mg) were mixed in methanol (5 mL) and stirred for 12 h. Then, ICG (1 mg) was added to the mixture under stirring. After 2 h, lauric acid (0.1 g) was added and the mixture was stirred for another 5 h. The precipitates were washed with methanol for three times. The as-obtained composites were denoted as (MSNs@CaO$_2$-ICG)@LA. The loading capacity of ICG was determined by dissolving the (MSNs@CaO$_2$-ICG)@LA in methanol, and the characteristic absorption at 788 nm was measured by UV-1800 spectrophotometer (Shimadzu, Japan).

**Extracellular measurement of H$_2$O$_2$ generation**. The H$_2$O$_2$ generation from CaO$_2$ was measured by Cu (II)-neocuproine spectrophotometric method[41]. In brief, 49 µL of PBS solution (10 mM, pH = 7.4, 6.5 or 5.5) of CaO$_2$ or CaO$_2$@LA ([CaO$_2$]=1 mg mL$^{-1}$, 1 µL), 25 µL of 0.01 M CuSO$_4$ and 25 µL of 0.01 M neocuproine solution were added to a 96-well plate sequentially. Then the 96-well plate was shaken at room temperature or at 46 °C. At appointed time, the absorbance at 450 nm was measured by a microplate reader. The standard curve of H$_2$O$_2$ was obtained by measuring the absorbance of a known concentration of H$_2$O$_2$ (0~1000 µM) in the same way and then the H$_2$O$_2$ concentration of samples could be calculated.

**Extracellular measurement of O$_2$ concentration**. Three mL of CaO$_2$ or CaO$_2$@LA methanol solution (1 mg mL$^{-1}$) were added to 27 mL PBS (10 mM, pH = 7.4) under vigorous stirring at room temperature or at 46 °C. Then we monitored the O$_2$ concentration of solution by a portable dissolved oxygen meter (JPBJ-608, Rex, INESA Scientific Instrument) in real time.

**Extracellular measurement of $^1$O$_2$ generation**. DPBF solution (10 µL, 10 mM in DMSO) was added to the sample solution (100 µg mL$^{-1}$, 2 mL) under irradiation (808 nm, 0.64 W cm$^{-2}$) and the absorbance of DPBF solution at 420 nm was recorded every minute.

**Extracellular measurement of photothermal property**. The temperature of different concentration of (MSNs@CaO$_2$-ICG)@LA (0, 25, 125 or 250 mg mL$^{-1}$, aqueous dispersion) was recorded by an OMEGA 4-channel datalogger thermometer under irradiation of 808 nm laser (0.64 W cm$^{-2}$, 10 min).

**Extracellular measurement of NIR-triggered ICG release**. The aqueous solution of (MSNs@CaO$_2$-ICG)@LA (200 µg mL$^{-1}$, 1 mL) was irradiated by laser (808 nm, 0.64 W cm$^{-2}$) with different time (0, 2, 4, 6, and 8 min) or was not exposed to the NIR laser. Then, the sample solution was centrifuged. The released ICG in supernatant solution was characterized by measuring the absorption of ICG at 788 nm.

**In vitro MRI imaging property**. In all, 100 µL MSNs or (MSNs@CaO$_2$-ICG)@LA ([Mn]: 5 mM) was added into 900 µL PBS solution (10 mM, pH = 7.4) containing 25 mM NaHCO$_3$/5% CO$_2$ without or with GSH (10 mM). After shaken at 37 °C for 1 h, MRI images and the T$_1$ relaxation time of diluted supernatant (Mn concentration: 0, 0.1, 0.2, 0.4, and 0.5 mM) were measured by MRI system.

**Cell cytotoxicity**. The human MCF-7 breast cancer cells, human A549 adenocarcinoma alveolar basal epithelial cells and NHDF normal human dermal fibroblasts cells were purchased from Cell Bank, the Committee of Type Culture Collection of Chinese Academy of Sciences. NHDF, MCF-7, and A549 cells were seeded in 96-well plates (10$^4$ cells per well) respectively and incubated in DMEM medium containing 10% FBS and 1% antibiotics (penicillin−streptomycin, 10,000 U mL$^{-1}$) at 37 °C under 5% CO$_2$ for 12 h. Then the cells were incubated with OPTI-MEM solution containing (MSNs@CaO$_2$-ICG)@LA at desired concentrations for 4 h, washed with PBS (10 mM, pH = 7.4) for three times and further incubated with DMEM medium for another 24 h. Finally, culture medium containing 10% of CCK-8 was added to each well. After incubation at 37 °C for 1 h, the absorbance at 450 nm of each well was obtained by microplate reader.

**Intracellular antitumor performance**. MCF-7 cells were seeded in 96-well plates (10$^4$ cells per well) for 12 h, the cells for hypoxia cytotoxicity evaluation were incubated in hypoxic chamber (1% O$_2$, 5% CO$_2$, and 94% N$_2$) for another 4 h. Then the cells were incubated with OPTI-MEM containing PBS, MSNs, (MSNs-ICG)@LA or (MSNs@CaO$_2$-ICG)@LA ([MSNs]=25 µg mL$^{-1}$) for 4 h. After replaced with fresh DMEM medium, the cells were irradiated with laser (808 nm, 0.64 W cm$^{-2}$) for 10 min and then further incubated for 24 h. For CCK-8 cytotoxicity assay, cells in each well were incubated with the culture medium containing 10% of CCK-8 for 1 h and measured by microplate reader.

**Intracellular hypoxia relief and ROS measurement**. Adherent MCF-7 cells were incubated in normoxia (5% O$_2$, 21% CO$_2$, and 74% N$_2$) or hypoxia chamber (1% O$_2$, 5% CO$_2$, and 94% N$_2$) for 4 h to build normoxic or hypoxic environment. Then the cells were incubated with OPTI-MEM solution containing PBS (10 mM,

pH = 7.4), MSNs, (MSNs-ICG)@LA or (MSNs@CaO$_2$-ICG)@LA for 4 h ([MSNs] = 25 µg mL$^{-1}$), washed with PBS (10 mM, pH = 7.4) and co-stained with 1 µL of DCFH-DA (10 mM) and 1 µL of hypoxia detection probe (1 mM) (Hypoxia Detection Kit, Enzo) for 15 min. After washed with PBS (10 mM, pH = 7.4) for three times, the cells were irradiated by a 808 nm laser (0.64 W cm$^{-2}$) for 10 min and observed by CLSM.

**In vivo antitumor performance**. MCF-7 cancer-bearing female Balb/c mice (4 weeks) were purchased from Beijing Vital River Laboratory Animal Technology Co., Ltd. and used in accordance with the guidelines of the Department of Laboratory Animal Science of Peking University Health Science Center. All animal experiments were conducted and agreed with the Institutional Animal Care and Use Committee of the Beijing Institute of Basic Medical Science (Beijing, China). The tumor-bearing mouse model was built by subcutaneous injection of MCF-7 cells ($2 \times 10^7$ mL$^{-1}$, 100 µL) into the right axilla of each mouse. The mice were randomly distributed into six groups for in vivo experiments (5 mice in each group) when the tumor volumes reached about 50 mm$^3$ and intravenously injected with different formulations ([MSNs]= 5 mg kg$^{-1}$): (1) PBS (control group); (2) 808 nm laser; (3) (MSNs@CaO$_2$-ICG)@LA; (4) PEG-modified MSNs; (5) (MSNs-ICG)@LA + 808 nm laser; (6) (MSNs@CaO$_2$-ICG)@LA + 808 nm laser. The 808 nm laser-irradiation was conducted after injection for 24 h (0.64 W cm$^{-2}$, 10 min). The tumor size ($V$) was calculated as follows: $V =$ width$^2 \times$ length/2 and measured every 2 days. After 2 weeks, tumors and main organs were collected from the killed mice for further analysis.

**In vivo blood circulation and biodistribution**. MCF-7 cancer-bearing female mice were intravenously injected with (MSNs@CaO$_2$-ICG)@LA (10 mg kg$^{-1}$). At indicated time points (0.1, 0.5, 1, 2, 4, 6, 11, 24 h), we collected 50 µL blood from the tail of each mouse. After intravenous injection for 24 h, the mice were killed to measure the Mn amount in liver, spleen, kidney, heart, lung, tumor as well as blood samples by ICP-OES.

**In vivo MRI imaging**. The in vivo MRI imaging experiments were conducted on MCF-7 tumor-bearing female mice when the tumor volume reached about 100 mm$^3$. MSNs or (MSNs@CaO$_2$-ICG)@LA ([MSNs] = 5 mg kg$^{-1}$) were intra-tumorally injected into the tumor sites. After 4 h, one of the (MSNs@CaO$_2$-ICG)@LA-injected mice were irradiated at the tumor sites by 808 nm-laser for 10 mim. After 12 h, T$_1$-weighted MR images were recorded with an animal MRI scanner (BioSpec70/20USR, Bruker, Germany).

**Statistical analysis**. Data were calculated and processed as mean ± SD. Comparison analysis between groups was conducted by student's $t$ test (two tailed).

**Reporting summary**. Further information on research design is available in the Nature Research Reporting Summary linked to this article.

## Data availability

All relevant data are available from the authors. The source data underlying Figs. 3a–c, 3f, 4a, c, f, 5a, b as well as Supplementary Figs 4, 5b, c, 6, 7, 9, 10, 16, and Supplementary Table 1 are provided as a Source Data file.

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

## Acknowledgements

The work was supported by National Natural Science Foundation of China (21874008, 21727815) and Special Foundation for State Major Research Program of China (2019YFC1606603), the work was also supported by Major Program of National Natural Science Foundation of China (21890740, 21890742); the Fundamental Research Funds for the Central Universities (FRF-TP-18-007B1, FRF-TP-17-050A1) and Beijing Municipal Science and Technology Commission (z131102002813058).

## Author contributions

C.L. and H.D. designed the present work. H.D. and X.Z. supervised the project. C.L. synthesized and characterized the nanomaterials, analyzed the data, and wrote the manuscript. C.L., D.W., and Y.CH. performed extracellular and intracellular experiments. C.L. and Y.CA. performed the animal experiments. T.X. and L.S. provided useful suggestions to this work. All the authors contributed to the discussion during the whole project.

## Competing interests

The authors declare no competing interests.
