## [Peer Review File · Nature Communications]

Reviewers' Comments:

Reviewer #1:

Remarks to the Author:

This paper is dedicated to the preparation of manganese silicate functionalized with CaO₂, ICG and lauric acid ((MSNs@CaO₂-ICG)@LA), used as thermoresponsive nanosystem for photodynamic/chemodynamic synergistic cancer therapy. The authors characterize their systems by various techniques, including TEM, XPS, absorption and in vivo/vitro experiments, and presents a detailed discussion of their experimental results.

The paper potentially contains some interesting data, but does not raise to the high standards of scientific level and broad readership request of a top-level journal like Nature Communications. Lots of works have been carried out so far on the use of smart nanosystem for synergistic cancer therapy for similar applications, and this one does not seem provide any novel conceptual leap/groundbreaking result that might justify its publication in this journal. The manuscript is in fact very technical and should be submitted to a specialized journal, after a careful consideration of all the following issues

(1) The Introduction does not proper evidence where the innovation and impact of the present manuscript lie with respect to the state-of-the-art in the field. This should be carefully put forward in detail.

(2) XPS analysis for the quantification of the metal ion is usually sensitive to the elements on the surface (~several nm). The accuracy for the quantification is largely limited when authors used XPS to characterize the MSN@CaO₂ with a diameter of 120 nm.

(3) On page 7 line 104, Figure 2D is not the TEM image. It should be Figure 2B.

(4) In general, the GSH concentrations from different tumor cells are different. Authors should take different GSH concentrations into account for the investigation of GSH effect on MSNs biodegradation.

(5) In Figure 3, authors would like to figure out the H₂O₂ and O₂ generation and thermo-responsive property. The uncertainties of the amount of the H₂O₂ and O₂ generation and the concentration of the nanoparticles should be included in Figure 3A and 3B. The response of (MSNs@CaO₂-ICG)@LA under irradiation to generate H₂O₂ and O₂ should be given. For the O₂ generation experiments, why the amount of the dissolved oxygen of CaO₂ is largely increased after coating with LA? Do the samples (CaO₂ and CaO₂@LA) have the same concentration? In Figure 3D, the Y-axis is the change of the absorbance intensity, and should be advised.

(6) In general, H₂O₂ will prefer to generate oxygen under Mn²⁺ catalysis. In Figure S9, authors used MB as the indicator to demonstrate the •OH generation, but MB is also used as the 1O₂ indicator. Author should include EPR testing to clarify the •OH generation. The samples for testing in Figure S9B and C should be (MSNs@CaO₂-ICG)@LA. Authors explained the MB degradation is decreased when further increased GSH concentration since excessive GSH would scavenge •OH conversely when GSH concentration is higher than 1 mM. The GSH concentration of the tumor cells can be 1-10 mM. Will •OH generated be scavenged? Whether this will affect the chemodynamic therapy results?

(7) ICG is the thermo-responsive molecule. The therapy results may be ascribed to the photothermal therapy from ICG. Authors should re-evaluate the therapy results.

Reviewer #2:

Remarks to the Author:

In this manuscript, the authors reported a self-supplying therapeutic platform that produces O₂

and H₂O₂ for hypoxia-relieved PDT, and induces GSH depletion to further improve ROS production efficiency. The system can initiate cascade reactions after 808 nm NIR light irradiation for the chemodynamic/photodynamic synergistic therapy. The technology is well designed. I would like to recommend the acceptance of the manuscript after proper revision. A list of more detailed suggestions are provided as follows:

- (1) The authors do not cite the extensive literature related to PDT-based combination therapy. The authors should highlight the novelty of their design compared to previous reports.
- (2) The authors claimed that the GSH induced degradation of MSNs and release of Mn ions. Why? In addition, more characterization about the stability of (MSNs@CaO₂-ICG)@LA in different mediums should be added.
- (3) The author claimed that the O₂ production could relieve hypoxia for enhanced PDT. Thus, ROS production and cell viability should be examined under hypoxia condition with different samples including (MSNs@CaO₂-ICG)@LA and (MSNs-ICG)@LA with or without irradiation.
- (4) Figure 2D demonstrated the higher fraction of Mn³⁺ ions compare to Mn²⁺ ions in the nanoparticles. Mn²⁺ ions are responsible for MRI, what is the role of Mn³⁺ in this design.
- (5) Please carefully check the serial number of figures, in line 104 the Fig. 2D should be Fig. 2B.
- (6) Scale bar is missing in Supplementary Fig. 11.

Reviewer #3:

Remarks to the Author:

In this manuscript, the authors described a H₂O₂/O₂ self-supplying agent (MSNs@CaO₂-ICG)@LA for hypoxia-relieved ICG-mediated PDT and H₂O₂-supplying MSN-based CDT. They found the nanoagents can effectively diminish tumor both in vitro and in vivo. In general, this MS provided a novel strategy for ROS generation enhancement in ROS-involved cancer therapies. However, the design of the nanoagent in this MS was similar with that published in ACS Nano 13,378 4267-4277 (2019). The authors should discuss the advantage of this study. Also, some others issues need to be addressed.

1. The stability of (MSNs@CaO₂-ICG)@LA should be detected.
2. In p7 line107, it should be figure 2B.
3. In figure 4A, the green fluorescence intensity in (MSNs@CaO₂-ICG)@LA group was similar with that in (MSNs-ICG)@LA group. The authors should optimized the experiments.
4. The specific anti-tumor effects of (MSNs@CaO₂-ICG)@LA was due to the tumor region irradiation of an 808 nm laser. The authors didn't mention a specific delivery of the carriers to tumor cells. Therefore, in figure 5B, the brightness of tumor site imaged by MRI can't reflect the actually tumor size, and irradiation sites affect the size of brightness.
5. The authors didn't show the uptake efficiency of (MSNs@CaO₂-ICG)@LA by tumor cells. Does the nanoagent function intracellularly or intercellularly in vivo? The authors should show the distribution of (MSNs@CaO₂-ICG)@LA in tumor site in vivo.
6. In figure 5, the authors should also detect the ROS, O₂ and H₂O₂ level, as well as Tunel+ cells in tumor tissues.
7. The biodistribution of (MSNs@CaO₂-ICG)@LA in blood circulation should be first measured.
8. In figure 3, the released Mn concentration and GSH/GSSG ratio should also be examined.
9. In figure 4, images of hypoxic marker should be stained to show the relief of oxygen by the nanosystem.

Dear Reviewers,

Thank you very much for the comments on the present work. Those comments are valuable and very helpful for revising and improving our paper, as well as the important guiding significance to our researches. In the following we make a point-to-point response to all the comments.

Detailed list of changes made and responses:

Response to Reviewer #1

(1) *The Introduction does not proper evidence where the innovation and impact of the present manuscript lie with respect to the state-of-the-art in the field. This should be carefully put forward in detail.*

Response: Thank you for your valuable advice. We have revised the Introduction to briefly but clearly exhibit the novelty in the manuscript as follow:

“.....However, the TME feature of hypoxia, depletable amount of H_2O_2 and the GSH depletion effect on ROS still limit ROS efficiency^{15,21,22}. Two different feasible strategies have been proposed to relieve hypoxia in PDT and supplement the cellular amount of H_2O_2 in CDT, respectively, amplifying endogenous O_2/H_2O_2 generation^{11,23-26} or directly delivering exogenous O_2/H_2O_2 into cells²⁷⁻²⁹. To date, there have indeed been some nanosystems for synergistic PDT/CDT, but most of them only overcome part of the limitations. For example, Liu et. al constructed sorafenib@ Fe^{3+} -tannic acid nanoparticles with GSH depletion property for PDT/CDT¹⁹. Copper ferrite nanosphere¹⁸, copper/manganese silicate nanospheres¹⁶ and ROS-activatable liposomes³⁰ have been reported for hypoxia-relieved and GSH-depleting synergistic PDT/CDT. At present, simultaneous hypoxia-relief, H_2O_2 -supplement and GSH-depletion nanosystems have been little reported, which is highly desirable in PDT/CDT combination therapy.

Herein, a H_2O_2/O_2 self-supplying thermoresponsive nanosystem, (MSNs@ CaO_2 -ICG)@LA, consisting of manganese silicate (MSNs) supported CaO_2 nanoparticles (NPs) and indocyanine green (ICG) with further surface coating of a

phase-change material lauric acid (LA, melting point: 44–46 °C)³¹, is reported for photodynamic/chemodynamic synergistic cancer therapy (**Fig. 1**). *CaO₂, a safe solid inorganic peroxide, can decompose to simultaneously release O₂ and H₂O₂ in contacting with water³² and have been widely applied in remediation of environmental contamination³³. In this nanosystem, the CaO₂ was protected from water by LA until the outer layer LA was melted due to the photothermal effect of ICG under the irradiation of a near-infrared (NIR) 808 nm laser. The exposed CaO₂ reacted with water to rapidly generate H₂O₂ and O₂, and accompanied exposure of inner MSNs. The released O₂ could relieve hypoxia for enhanced ICG-mediated PDT. The interaction between MSNs and glutathione (GSH) led to release of Fenton-like agent Mn²⁺ for H₂O₂-supplementing CDT and magnetic resonance imaging (MRI)-guided therapy. This GSH depletion further enhanced the ROS generation efficiency. Thus we first reported a smart system (MSNs@CaO₂-ICG)@LA can simultaneously overcome the main limitations including hypoxia, depletable amount of H₂O₂ and GSH elimination effect on ROS for synergistic PDT/CDT, and this “open source and reduce expenditure” ROS-produced way obtained excellent tumor inhibition effect both in vitro and in vivo, provide a universal idea of therapeutic nanoagents design for synergistic PDT/CDT.”*

(2) XPS analysis for the quantification of the metal ion is usually sensitive to the elements on the surface (~several nm). The accuracy for the quantification is largely limited when authors used XPS to characterize the MSN@CaO₂ with a diameter of 120 nm.

Response: Thank you for your kind remind. In this paper, XPS analysis was conducted to evaluate the proportion of different valence state of Mn rather than quantification of Mn ions. Similar XPS analysis was also performed in previous papers (J. Am. Chem. Soc. **2016**, 138 (31), 9881-94; Adv. Healthc. Mater. 2017, 6 (22)) to analyze the valence state of Mn or Fe in the mesoporous silica-based nanoparticles with a diameter of ~100 nm. As for the quantification of Mn ion, we have performed

the ICP measurements to quantify the Mn species in MSNs which was about 3.83% (mass ratio).

(3) *On page 7 line 104, Figure 2D is not the TEM image. It should be Figure 2B.*

Response: Thank you for your valuable advice. We have revised the figure number in the manuscript as follows.

“The TEM image of (MSNs@CaO₂-ICG)@LA showed the uniform size (Fig. 2B) and the change of surface zeta potential (Supplementary Fig. 4) confirmed the preparation process.”

(4) *In general, the GSH concentrations from different tumor cells are different. Authors should take different GSH concentrations into account for the investigation of GSH effect on MSNs biodegradation.*

Response: Thank you for your valuable advice. We have added the TEM images of MSNs biodegradation in Supplementary Fig. 3 when GSH concentrations is 1 mM.

Supplementary Fig. 3 The effect of GSH (0, 1, 10 mM) on biodegradation of MSNs (100 µg/mL) in PBS (10 mM, pH 7.4) for various periods of time. Scale bar: 100 nm.

(5) *In Figure 3, authors would like to figure out the H₂O₂ and O₂ generation and thermo-responsive property. The uncertainties of the amount of the H₂O₂ and O₂*

generation and the concentration of the nanoparticles should be included in Figure 3A and 3B. The response of (MSNs@CaO₂-ICG)@LA under irradiation to generate H₂O₂ and O₂ should be given. For the O₂ generation experiments, why the amount of the dissolved oxygen of CaO₂ is largely increased after coating with LA? Do the samples (CaO₂ and CaO₂@LA) have the same concentration? In Figure 3D, the Y-axis is the change of the absorbance intensity, and should be advised.

Response: We have revised the caption of Figure 3A, B in the manuscript and the Y-axis of Figure 3D. In Figure 3A, B, we aimed to test the H₂O₂ and O₂ generation property of CaO₂. Considering that the heat and ¹O₂ generated by ICG under laser irradiation will cause H₂O₂ decomposition and O₂ consumption respectively and affect the final testing results, we used CaO₂@LA rather than (MSNs@CaO₂-ICG)@LA to evaluate the H₂O₂ and O₂ generation property of CaO₂. The dissolved oxygen concentration of CaO₂@LA at 25 °C was increased a little bit but not much comparing with that of CaO₂ at 25 °C and CaO₂@LA at 46 °C, which may be caused by the dissolved oxygen in the added solution. The sample CaO₂ and CaO₂@LA have the same concentration of CaO₂ in the same set of experiments.

Fig. 3 (A) H₂O₂ cumulative release profile ([CaO₂]=10 μg/mL) and (B) O₂ concentration measurement in PBS (10 mM, pH=7.4) ([CaO₂]=100 μg/mL). (C) Time-dependent degradation of DPBF irradiated by laser for 10 min. ([MSNs]=50 μg/mL, [ICG]=8 μg/mL. Laser: 808 nm, 0.64 W/cm²). (D) EPR analysis of •OH production in each group with different treatment. DMPO was used as the spin-trapping agent. The sample in each group was (MSNs@CaO₂-ICG)@LA aqueous solution ([MSNs] = 50 μg/mL). (E) In vitro

MRI of different solution and (F) the corresponding r_1 value ([MSNs] = 50 $\mu\text{g/mL}$. Laser: 808 nm, 0.64 W/cm^2 , 10 min).

(6) In general, H_2O_2 will prefer to generate oxygen under Mn^{2+} catalysis. In Figure S9, authors used MB as the indicator to demonstrate the $\bullet\text{OH}$ generation, but MB is also used as the $^1\text{O}_2$ indicator. Author should include EPR testing to clarify the $\bullet\text{OH}$ generation. The samples for testing in Figure S9B and C should be (MSNs@ CaO_2 -ICG)@LA. Authors explained the MB degradation is decreased when further increased GSH concentration since excessive GSH would scavenge $\bullet\text{OH}$ conversely when GSH concentration is higher than 1 mM. The GSH concentration of the tumor cells can be 1-10 mM. Will $\bullet\text{OH}$ generated be scavenged? Whether this will affect the chemodynamic therapy results?

Response: Actually, it has been reported that Mn^{2+} can triggered $\bullet\text{OH}$ production from H_2O_2 in Fenton reaction with the help of bicarbonate (HCO_3^-) which is sufficient in vivo as one of the most important physiological buffers (Angew. Chem. Int. Ed. 2018, 57 (18), 4902-4906; ACS Nano 2019, 13 (12), 13985-13994; New J. Chem., 2009, 33, 34), and we also proved it in this paper that a significant decrease in absorbance was observed when MB was incubated with H_2O_2 , Mn^{2+} and NaHCO_3 buffer, whereas no apparent change in the MB absorbance was detected after the same treatment in aqueous solution (see Supplementary Fig. 10A).

Supplementary Fig. 10 (A) UV-vis absorption spectra of MB degradation in different solutions [Mn] = 0.5 mM, [H_2O_2] = 8 mM, [$\text{NaHCO}_3/5\% \text{CO}_2$] = 25 mM). (B) MB degradation by $\bullet\text{OH}$ generated from different concentration of GSH-treated MSNs (100 $\mu\text{g/mL}$) plus H_2O_2 (8 mM) and (C) different concentration of H_2O_2 -treated MSNs (100 $\mu\text{g/mL}$) plus GSH (1 mM). [$\text{NaHCO}_3/5\% \text{CO}_2$] = 25 mM.

In addition, MB was used as $\bullet\text{OH}$ indicator in many previous literature (Angew. Chem. Int. Ed. 2018, 57 (18); ACS Nano 2018, 12 (5), 4886-4893; J. Am. Chem. Soc.

2018, 141 (2), 849-857). In our paper, MB degradation experiments were conducted without NIR laser irradiation, thus the results will not be affected by $^1\text{O}_2$ even $^1\text{O}_2$ can cause MB degradation. We also added EPR testing to clarify the $\cdot\text{OH}$ generation as shown in **Fig. 3D** and revised the manuscript as follow:

“.....The H_2O_2 concentration-dependent Fenton-like effect provided the great possibility of enhanced $\cdot\text{OH}$ generation by H_2O_2 self-supply from CaO_2 (Supplementary Fig. 10C). Similar results to the MB degradation experiments were also obtained by electron paramagnetic resonance (EPR) analysis of $\cdot\text{OH}$ production of $(\text{MSNs}@/\text{CaO}_2\text{-ICG})@/\text{LA}$ as shown in **Fig. 3D**.”

Fig. 3 (A) H_2O_2 cumulative release profile ($[\text{CaO}_2]=10 \mu\text{g}/\text{mL}$) and (B) O_2 concentration measurement in PBS (10 mM, pH=7.4) ($[\text{CaO}_2]=100 \mu\text{g}/\text{mL}$). (C) Time-dependent degradation of DPBF irradiated by laser for 10 min. ($[\text{MSNs}]=50 \mu\text{g}/\text{mL}$, $[\text{ICG}]=8 \mu\text{g}/\text{mL}$. Laser: 808 nm, $0.64 \text{ W}/\text{cm}^2$). (D) EPR analysis of $\cdot\text{OH}$ production in each group with different treatment. DMPO was used as the spin-trapping agent. The sample in each group was $(\text{MSNs}@/\text{CaO}_2\text{-ICG})@/\text{LA}$ aqueous solution ($[\text{MSNs}] = 50 \mu\text{g}/\text{mL}$). (E) In vitro MRI of different solution and (F) the corresponding r_1 value ($[\text{MSNs}] = 50 \mu\text{g}/\text{mL}$. Laser: 808 nm, $0.64 \text{ W}/\text{cm}^2$, 10 min).

The testing in Figure S9B and C (now as Figure S10B and C) aim to study the relationship between GSH depletion-enhanced $\cdot\text{OH}$ generation of MSNs in CDT and GSH/ H_2O_2 concentration. Review think the sample is supposed to be $(\text{MSNs}@/\text{CaO}_2\text{-ICG})@/\text{LA}$, in that case the $^1\text{O}_2$ and heat caused by ICG under NIR laser irradiation may affect MB degradation. Therefore, we choose the MSNs as samples in Figure S9B and C. The generated $\cdot\text{OH}$ can be scavenged by GSH, and this

will affect the chemodynamic therapy results, so that's why we used MSNs to scavenge GSH for better therapeutic effect.

(7) ICG is the thermo-responsive molecule. The therapy results may be ascribed to the photothermal therapy from ICG. Authors should re-evaluate the therapy results.

Response: Thank you for your comment. Generally, if quantifying the thermal exposure as a thermal dose in equivalent minutes at 43 °C in PTT, thermal doses of 120-240 min at 43 °C generate considerable tissue necrosis and 240-540 min is lethal thermal dose exposure. (Int. J. Hyperthermia 2005, 21, 745-53). To achieve efficient tumor-tissue ablation, a high temperature of over 50 °C is required to overcome the thermoresistance caused by heat shock protein (ACS Nano 2019, 13(2), 1499-1510). In previous literature, ICG has been used to generate ¹O₂ (ACS Nano 2014, 8 (11), 11529-42; Biomaterials 2013, 34 (33), 8314-22.) or heat (ACS Nano 2013, 7 (3), 2056-67) under 808 nm laser-irradiation. However, only when maximum temperature up to 55 °C for 5 min complete tumor remission could be achieved in ICG-mediated PTT (ACS Nano 2013, 7 (3), 2056-67; ACS Nano 2016, 10 (11), 10049-10057). In our work, we utilized the heat generating from ICG to melt the thermo-responsive LA. We measured the temperature changes at the tumor site of the PBS or (MSNs@CaO₂-ICG)@LA treated-mice under irradiation ([MSNs]= 5 mg/kg; laser: 808 nm, 0.64 W cm⁻², 10 min) by IR thermal imaging as shown in Supplementary Fig. 13. Comparing with the PBS-treated group, the tumor-site temperature of (MSNs@CaO₂-ICG)@LA-treated mice had obviously change and gradually increased to 46.8 °C, which exceeded the melting point temperature of LA but was not enough to harm the skin and tissue. Therefore, in our work, ICG is mainly used to produce ROS in PDT and generate mild heat to melt LA, while photothermal therapy from ICG on the therapeutic results is negligible.

Supplementary fig. 13 IR thermal images of the mcf-7 tumor-bearing mice in groups of pbs- or (msns@cao₂-icg)@la treated-mice under irradiation (([msns]= 5 mg/kg; laser: 808 nm, 0.64 w cm⁻², 10 min).

Response to Reviewer #2

(1) *The authors do not cite the extensive literature related to PDT-based combination therapy. The authors should highlight the novelty of their design compared to previous reports.*

Response: Thank you for your valuable advice. We have added the extensive literature related to PDT-based combination therapy and revised the introduction part to briefly but clearly exhibit the novelty.

“.....On this basis, ROS-mediated therapies, such as photodynamic therapy (PDT)³⁻⁶ and chemodynamic therapy (CDT)⁷⁻¹¹, are developed to disrupt the cellular self-adaptation mechanisms and induce cell death based on ROS-generating agents¹².”

“.....However, the TME feature of hypoxia, depletable amount of H₂O₂ and the GSH depletion effect on ROS still limit ROS efficiency^{15,21,22}. Two different feasible strategies have been proposed to relieve hypoxia in PDT and supplement the cellular amount of H₂O₂ in CDT, respectively, amplifying endogenous O₂/H₂O₂ generation^{11,23-26} or directly delivering exogenous O₂/H₂O₂ into cells²⁷⁻²⁹. To date, there have indeed been some nanosystems for synergistic PDT/CDT, but most of them only overcome part of the limitations. For example, Liu et. al constructed sorafenib@Fe³⁺-tannic acid nanoparticles with GSH depletion property for

PDT/CDT¹⁹. Copper ferrite nanosphere¹⁸, copper/manganese silicate nanospheres¹⁶ and ROS-activatable liposomes³⁰ have been reported for hypoxia-relieved and GSH-depleting synergistic PDT/CDT. At present, simultaneous hypoxia-relief, H₂O₂-supplement and GSH-depletion nanosystems have been little reported, which is highly desirable in PDT/CDT combination therapy.

Herein, a H₂O₂/O₂ self-supplying thermoresponsive nanosystem, (MSNs@CaO₂-ICG)@LA, consisting of manganese silicate (MSNs) supported CaO₂ nanoparticles (NPs) and indocyanine green (ICG) with further surface coating of a phase-change material lauric acid (LA, melting point: 44~46 °C)³¹, is reported for photodynamic/chemodynamic synergistic cancer therapy (Fig. 1). CaO₂, a safe solid inorganic peroxide, can decompose to simultaneously release O₂ and H₂O₂ in contacting with water³² and have been widely applied in remediation of environmental contamination³³. In this nanosystem, the CaO₂ was protected from water by LA until the outer layer LA was melted due to the photothermal effect of ICG under the irradiation of a near-infrared (NIR) 808 nm laser. The exposed CaO₂ reacted with water to rapidly generate H₂O₂ and O₂, and accompanied exposure of inner MSNs. The released O₂ could relieve hypoxia for enhanced ICG-mediated PDT. The interaction between MSNs and glutathione (GSH) led to release of Fenton-like agent Mn²⁺ for H₂O₂-supplementing CDT and magnetic resonance imaging (MRI)-guided therapy. This GSH depletion further enhanced the ROS generation efficiency. Thus we first reported a smart system (MSNs@CaO₂-ICG)@LA can simultaneously overcome the main limitations including hypoxia, depletable amount of H₂O₂ and GSH elimination effect on ROS for synergistic PDT/CDT, and this “open source and reduce expenditure” ROS-produced way obtained excellent tumor inhibition effect both in vitro and in vivo, provide a universal idea of therapeutic nanoagents design for synergistic PDT/CDT.”

Added Reference:

3. Li, Y., *et al.* Heterodimers Made of Upconversion Nanoparticles and Metal–Organic Frameworks. *J. Am. Chem. Soc.* **139**, 13804–13810 (2017).

4. Yu, Z., Zhou, P., Pan, W., Li, N. & Tang, B. A biomimetic nanoreactor for synergistic chemiexcited photodynamic therapy and starvation therapy against tumor metastasis. *Nat. Commun.* **9**, 5044 (2018).
5. Yang, G., *et al.* Hollow MnO₂ as a tumor-microenvironment-responsive biodegradable nano-platform for combination therapy favoring antitumor immune responses. *Nat. Commun.* **8**, 902 (2017).
6. Liu, C., *et al.* Nd³⁺-Sensitized Upconversion Metal-Organic Frameworks for Mitochondria-Targeted Amplified Photodynamic Therapy. *Angewandte Chemie (International ed. in English)* (2019).
7. Zhang, C., *et al.* Synthesis of Iron Nanometallic Glasses and Their Application in Cancer Therapy by a Localized Fenton Reaction. *Angew. Chem. Int. Ed.* **55**, 2101-2106 (2016).
8. Huo, M., Wang, L., Chen, Y. & Shi, J. Tumor-selective catalytic nanomedicine by nanocatalyst delivery. *Nat. Commun.* **8**, 357 (2017).
9. Lin, L.S., *et al.* Simultaneous Fenton-like Ion Delivery and Glutathione Depletion by MnO₂-Based Nanoagent to Enhance Chemodynamic Therapy. *Angew. Chem. Int. Ed.* **57**, 4902-4906 (2018).
10. Ma, B., *et al.* Self-Assembled Copper-Amino Acid Nanoparticles for In Situ Glutathione "AND" H₂O₂ Sequentially Triggered Chemodynamic Therapy. *J. Am. Chem. Soc.* **141**, 849-857 (2018).
11. Lin, L.S., *et al.* Synthesis of Copper Peroxide Nanodots for H₂O₂ Self-Supplying Chemodynamic Therapy. *J. Am. Chem. Soc.* **141**, 9937-9945 (2019).
22. Gong, N., *et al.* Carbon-dot-supported atomically dispersed gold as a mitochondrial oxidative stress amplifier for cancer treatment. *Nat. Nanotechnol.* **14**, 379-387 (2019).
30. Zhao, Z., *et al.* Reactive Oxygen Species-Activatable Liposomes Regulating Hypoxic Tumor Microenvironment for Synergistic Photo/Chemodynamic Therapies. *Adv. Funct. Mater.* **29**, 1905013 (2019).

(2) The authors claimed that the GSH induced degradation of MSNs and release of Mn ions. Why? In addition, more characterization about the stability of (MSNs@CaO₂-ICG)@LA in different mediums should be added.

Response: Thank you for your valuable comments. Generally, it is difficult for the –Si–O–Si– framework of mesoporous silica to biodegrade in physiological conditions (J. Am. Chem. Soc. 2012, 134, 13997; Nanoscale 2015, 7, 2676). It has been reported that metal (M) doped in mesoporous silica can substitute Si to form M–O–Si bonds within the framework and especially the Mn–O bonds are sensitive to reducing microenvironment. The reduction of Mn species in MSNs by GSH may leave the silica framework with abundant defects and reactive sites, which further accelerate the biodegradation of silica framework to release Mn. (J. Am. Chem. Soc. 2016, 138, 9881; Adv. Healthc. Mater. 2017, 6). As for the stability of (MSNs@CaO₂-ICG)@LA in different mediums, we tested the DLS size and zeta potential in different mediums as shown in Supplementary Fig. 5.

Supplementary Fig. 5 (A) DLS characterization of (MSNs@CaO₂-ICG)@LA in water. (B) DLS characterization and (C) surface ζ potential of (MSNs@CaO₂-ICG)@LA in different mediums overnight.

(3) The author claimed that the O₂ production could relieve hypoxia for enhanced PDT. Thus, ROS production and cell viability should be examined under hypoxia condition with different samples including (MSNs@CaO₂-ICG)@LA and (MSNs-ICG)@LA with or without irradiation.

Response: Thank you for your valuable comments. We have added the measurement of ROS production and cell viability under hypoxia condition as shown in Fig. 4F and Supplementary Fig. 12, and revised the relevant part in manuscript as follow:

Fig. 4 (A) Cell viability of MCF-7, A549 and NHDF cells treated with different concentrations of (MSNs@CaO₂-ICG)@LA by CCK-8 assay. (B) The rate of MCF-7 cells uptaking (MSNs@CaO₂-ICG)@LA ([MSNs]= 25 µg/mL) and (C) corresponding mean fluorescence intensity by flow cytometry after different incubation time. (D) Fluorescence images showing cellular uptake of (MSNs@CaO₂-ICG)@LA ([MSNs]= 25 µg/mL) in MCF-7 cells after incubation for 4 h. Scale bar: 20 µm. (E) Fluorescence images showing ROS and hypoxia level in MCF-7 cells with different treatment under normoxia condition. ([MSNs]=25 µg/mL. Laser: 0.64 W/cm², 10 min). Scale bar: 50 µm. (F) Cell viability of MCF-7 cells with different treatments under hypoxia or normoxia condition ([MSNs]=25 µg/mL. Laser: 0.64 W/cm², 10 min) (**p < 0.01 and ***p < 0.001, two-tailed t test). (G) Fluorescence images of Calcein-AM- and propidium iodide (PI)-costained MCF-7 cells with different treatments under normoxia condition ([MSNs]=25 µg/mL. Laser: 0.64 W/cm², 10 min). Scale bar: 200 µm.

Supplementary Fig. 12 Fluorescence images showing ROS and hypoxia level in MCF-7 cells with different treatment under hypoxia condition. ([MSNs]=25 $\mu\text{g/mL}$. Laser: 0.64 W/cm^2 , 10 min). Scale bar: 50 μm .

“.....As expected, the intracellular fluorescence imaging by ROS probe (DCFH-DA, green) and hypoxia probe demonstrated that the (MSNs@CaO₂-ICG)@LA-treated cells with irradiation displayed the strongest green fluorescence and slight enhanced red fluorescence compared with other groups both under normoxia or hypoxia conditions (Fig. 4E and Supplementary Fig. 12), indicating tremendous ROS generation enhanced by O₂ and H₂O₂ self-supply from CaO₂. In order to further investigate the enhanced CDT/PDT therapeutic effect, we used MCF-7 cancer cells as model cell to examine the anticancer effect of (MSNs@CaO₂-ICG)@LA in normoxia as well as hypoxia environment as contrast. Upon a 808 nm laser irradiation, the (MSNs@CaO₂-ICG)@LA treatment showed greatly enhanced cytotoxicity than both MSNs and (MSNs-ICG)@LA in normoxia environment (Fig. 4F), in consistent with the calcein-AM and propidium iodide (PI) co-staining results that most of cancer cells were killed when treated with (MSNs@CaO₂-ICG)@LA plus irradiation (Fig. 4G). Notably, the hypoxia condition had obvious influence on the cell killing ability of (MSNs-ICG)@LA-treated group, while that of (MSNs@CaO₂-ICG)@LA-treated group almost the same both in normoxia and in hypoxia conditions owing to the O₂ self-supplying property of CaO₂ (Fig. 4F).”

“Intracellular antitumor performance. MCF-7 cells were seeded in 96-well plates (10^4 cells/well) for 12 h, the cells for hypoxia cytotoxicity evaluation were incubated in hypoxic chamber (1% O₂, 5% CO₂, and 94% N₂) for another 4 h. Then the cells were incubated with OPTI-MEM containing PBS, MSNs, (MSNs-ICG)@LA or (MSNs@CaO₂-ICG)@LA ([MSNs]=25 µg/mL) for 4 h. After replaced with fresh DMEM medium, the cells were irradiated with laser (808 nm, 0.64 W/cm²) for 10 min and then further incubated for 24 h. For CCK-8 cytotoxicity assay, cells in each well were incubated with the culture medium containing 10% of CCK-8 for 1 h and measured by microplate reader.

Intracellular hypoxia relief and ROS measurement. Adherent MCF-7 cells were incubated in normoxia (5% O₂, 21% CO₂, and 74% N₂) or hypoxia chamber (1% O₂, 5% CO₂, and 94% N₂) for 4 h to build normoxic or hypoxic environment. Then the cells were incubated with OPTI-MEM solution containing PBS (10 mM, pH = 7.4), MSNs, (MSNs-ICG)@LA or (MSNs@CaO₂-ICG)@LA for 4 h ([MSNs] = 25 µg/mL), washed with PBS (10 mM, pH = 7.4) and co-stained with 1 µL of DCFH-DA (10 mM) and 1 µL of hypoxia detection probe (1 mM) (Hypoxia Detection Kit, Enzo) for 15 min. After washed with PBS (10 mM, pH = 7.4) for three times, the cells were irradiated by a 808 nm laser (0.64 W/cm²) for 10 min and observed by CLSM.”

(4) Figure 2D demonstrated the higher fraction of Mn³⁺ ions compare to Mn²⁺ ions in the nanoparticles. Mn²⁺ ions are responsible for MRI, what is the role of Mn³⁺ in this design.

Response: Thank you for your valuable comments. The oxidizing Mn³⁺ can be reduced by GSH to release Mn²⁺ ions as Fenton-like agent and MRI contrast agent, which will leave the silica framework with abundant defects and reactive sites and lead to the further biodegradation of silica framework.

(5) Please carefully check the serial number of figures, in line 104 the Fig. 2D should be Fig. 2B.

Response: Thank you for your valuable advice. We have revised the figure number in the manuscript as follows:

“The TEM image of (MSNs@CaO₂-ICG)@LA showed the uniform size (Fig. 2B) and the change of surface zeta potential (Supplementary Fig. 4) confirmed the preparation process.”

(6) Scale bar is missing in Supplementary Fig. 11.

Response: Thank you for your valuable advice. We have added the scale bar as follows:

Supplementary Fig. 14 H&E-stained images of organs obtained from mice in group 1-6 corresponding to Fig. 5E at 14 days. Scale bar: 100 μ m.

Response to Reviewer #3

In this manuscript, the authors described a H₂O₂/O₂ self-supplying agent (MSNs@CaO₂-ICG)@LA for hypoxia-relieved ICG-mediated PDT and H₂O₂-supplying MSN-based CDT. They found the nanoagents can effectively diminish tumor both in vitro and in vivo. In general, this MS provided a novel strategy for ROS generation enhancement in ROS-involved cancer therapies. However, the design of the nanoagent in this MS was similar with that published in

ACS Nano 13,378 4267-4277 (2019). The authors should discuss the advantage of this study. Also, some others issues need to be addressed.

Response: Thank you for your advice. The paper published in ACS Nano reported a copper/manganese silicate nanospheres (CMSNs). Although this nanosystem overcame the hypoxia and GSH scavenging effect on ROS, the therapeutic effect was severely limited by depletable endogenous H₂O₂, because the O₂ was produced from endogenous H₂O₂ catalyzed by CMSNs and •OH was produced from H₂O₂ in Fenton reaction catalyzed by Mn²⁺ released from CMSNs. While our nanosystem (MSNs@CaO₂-ICG)@LA in this paper utilized a safe solid inorganic peroxide, CaO₂, to simultaneously decompose and release O₂ and H₂O₂ in contacting with water, which break the limit of insufficient H₂O₂ in cancer cells. The CaO₂ was protected from water by LA until the outer layer LA was melted due to the photothermal effect of ICG under the irradiation of a near-infrared 808 nm laser. The interaction between MSNs and glutathione (GSH) led to release of Fenton-like agent Mn²⁺ for H₂O₂-supplementing CDT and magnetic resonance imaging-guided therapy. Therefore, our nanosystem in this paper was more intelligent than that published in ACS Nano 13,378 4267-4277 (2019).

(1) *The stability of (MSNs@CaO₂-ICG)@LA should be detected.*

Response: Thank you for your advice. We tested the DLS size and zeta potential in different mediums to evaluate the stability of (MSNs@CaO₂-ICG)@LA as shown in Supplementary Fig. 5.

Supplementary Fig. 5 (A) DLS characterization of (MSNs@CaO₂-ICG)@LA in water. (B) DLS characterization and (C) surface ζ potential of (MSNs@CaO₂-ICG)@LA in different mediums overnight.

(2) In p7 line107, it should be figure 2B.

Response: Thank you for your valuable advice. We have revised the figure number in the manuscript as follows:

“The TEM image of (MSNs@CaO₂-ICG)@LA showed the uniform size (Fig. 2B) and the change of surface zeta potential (Supplementary Fig. 4) confirmed the preparation process.”

(3) In figure 4A, the green fluorescence intensity in (MSNs@CaO₂-ICG)@LA group was similar with that in (MSNs-ICG)@LA group. The authors should optimize the experiments.

Response: Thank you for your comment. In figure 4A, we showed the fluorescence images of cells and corresponding surface plot images about the distribution and intensity of green fluorescence, in which the green fluorescence intensity of (MSNs-ICG)@LA+808 nm group was obviously stronger than that of (MSNs@CaO₂-ICG)@LA group.

(4) The specific anti-tumor effects of (MSNs@CaO₂-ICG)@LA was due to the tumor region irradiation of an 808 nm laser. The authors didn't mention a specific delivery of the carriers to tumor cells. Therefore, in figure 5B, the brightness of tumor site imaged by MRI can't reflect the actually tumor size, and irradiation sites affect the size of brightness.

Response: Thank you for your comment. We performed the in vivo MRI by intratumorally injecting the samples into the tumor of the mice when the tumor volumes reached about 100 mm³, which have been adopted for most of non-specific-delivered nanomaterials in many previous papers (Adv. Funct. Mater.

2017, 1700626; Adv. Mater. 2017, 29 (47), 1701683; ACS Nano 2019, 13 (12), 13985-13994). In Figure 5B, the brightness of tumor sites indicated that the MSN was reduced by GSH to release Mn^{2+} because free Mn ions were easier to proceed chemical exchange with protons than isolated Mn centers in MSNs for enhanced T_1 signal (ACS Nano 2018, 12 (11), 11000-11012), so the 808 nm laser-irradiation to melt the LA was necessary for (MSNs@CaO₂-ICG)@LA to generate brightness at tumor sites. Therefore, the brightness of tumor site imaged by MRI was to monitor the therapy process rather than reflect the actual tumor size. As for the irradiation sites, we used laser vertical collimator to make the whole tumor to be irradiated, by which avoided affecting the size of brightness because of the irradiation sites. We have added the experiment process of in vivo MRI imaging in revised manuscript as follow:

“In vivo MRI imaging. The in vivo MRI imaging experiments were conducted on MCF-7 tumor-bearing femice when the tumor volume reached about 100 mm³. MSNs or (MSNs@CaO₂-ICG)@LA ([MSNs] = 5 mg/kg) were intratumorally injected into the tumor sites. After 4 h, one of the (MSNs@CaO₂-ICG)@LA-injected mice were irradiated at the tumor sites by 808 nm-laser for 10 min. After 12 h, T₁-weighted MR images were recorded with an animal MRI scanner (BioSpec70/20USR, Bruker, Germany).”

(5) *The authors didn't show the uptake efficiency of (MSNs@CaO₂-ICG)@LA by tumor cells. Does the nanoagent function intracellularly or intercellularly in vivo? The authors should show the distribution of (MSNs@CaO₂-ICG)@LA in tumor site in vivo.*

Response: Thank you for your comment. The nanoagent function intracellularly because it entered cells via an endolysosomal pathway and degraded by intracellular overexpressed GSH. We tested cell uptake rate of (MSNs@CaO₂-ICG)@LA in MCF-7 cells by flow cytometry after different incubation time and showed the location of (MSNs@CaO₂-ICG)@LA in MCF-7 cells by fluorescence imaging, the figures were shown as **Fig. 4B-D** as follow. It could be observed that after incubation

for 4 h most of (MSNs@CaO₂-ICG)@LA entered MCF-7 cells and located at lysosome. We also tested the GSH/GSSG ratio in MCF-7 cells before and after incubation with MSNs (50 µg/mL) for 12 h using GSH and GSSG Assay Kit (Beyotime), which was 12.20 and 7.57 respectively. The decreased GSH/GSSG ratio indicated the MSNs intracellularly functioned with excessive GSH. We revised the relevant part in manuscript as follow:

*“**Intracellular uptake of (MSNs@CaO₂-ICG)@LA.** Before evaluating the feasibility of (MSNs@CaO₂-ICG)@LA for in vivo antitumor therapy, the cytotoxicity and cell uptake of (MSNs@CaO₂-ICG)@LA were first investigated. As shown in **Fig. 4A**, the (MSNs@CaO₂-ICG)@LA showed little cytotoxicity toward MCF-7, A549 and NHDF cells when the concentration was from 0 to 50 µg/mL after incubation for 12 h, indicating good biocompatibility. The flow cytometry was conducted to measure the fluorescence intensity of ICG in MCF-7 cells treated with (MSNs@CaO₂-ICG)@LA at different incubation time point (**Fig. 4B**), and the corresponding analysis of mean fluorescence intensity was shown in **Fig. 4C**. Comparing with the blank control group, the cells treated with (MSNs@CaO₂-ICG)@LA demonstrated high uptake rate as the incubation time extended, and incubation for 4 h was sufficient. **Fig. 4D** demonstrated that the location of (MSNs@CaO₂-ICG)@LA in MCF-7 cells overlapped with lysosome after incubation for 4 h, suggesting the endolysosomal pathway.”*

Fig. 4 (A) Cell viability of MCF-7, A549 and NHDF cells treated with different concentrations of (MSNs@CaO₂-ICG)@LA by CCK-8 assay. (B) The rate of MCF-7 cells uptaking (MSNs@CaO₂-ICG)@LA ([MSNs]= 25 μg/mL) and (C) corresponding mean fluorescence intensity by flow cytometry after different incubation time. (D) Fluorescence images showing cellular uptake of (MSNs@CaO₂-ICG)@LA ([MSNs]= 25 μg/mL) in MCF-7 cells after incubation for 4 h. Scale bar: 20 μm. (E) Fluorescence images showing ROS and hypoxia level in MCF-7 cells with different treatment under normoxia condition. ([MSNs]=25 μg/mL. Laser: 0.64 W/cm², 10 min). Scale bar: 50 μm. (F) Cell viability of MCF-7 cells with different treatments under hypoxia or normoxia condition ([MSNs]=25 μg/mL. Laser: 0.64 W/cm², 10 min) (**p < 0.01 and ***p < 0.001, two-tailed t test). (G) Fluorescence images of Calcein-AM- and propidium iodide (PI)-costained MCF-7 cells with different treatments under normoxia condition ([MSNs]=25 μg/mL. Laser: 0.64 W/cm², 10 min). Scale bar: 200 μm.

(6) In figure 5, the authors should also detect the ROS, O₂ and H₂O₂ level, as well as Tunel+ cells in tumor tissues.

Response: Thank you for your advice. We have detected the ROS and O₂ level and in tumor tissues as well as TUNEL-stained tumor slices added as Fig. 5G, Supplementary Fig. 15 and Fig. 5H respectively. As for the H₂O₂ level in tumor

tissues, it was hard for us to buy suitable probes to distinguish H_2O_2 from other ROS species. We have revised the manuscript as follow:

Fig. 5 (A) Blood circulation and (B) biodistribution of (MSNs@CaO₂-ICG)@LA by measuring Mn concentrations over a span of 24 h after intravenous injection. (C) In vivo MRI of MCF-7 tumor-bearing mice with different treatments after 24 h (Laser: 808 nm, 0.64 W/cm², 10 min). (D) The body weight and (E) the relative cancer volume changes of MCF-7 tumor-bearing mice after various treatments in 14 days (**p < 0.01 and ***p < 0.001, two-tailed t test). (F) Representative photos of the mice with different treatments and corresponding tumor tissues collected from different groups at 14 days. (G) DCFH (green)- and DAPI (blue)-contained tumor tissues collected from different groups at 24 h post-injection. Scale bar: 200 μm. (H) H&E- or TUNEL-stained tumor slices collected from different groups at 14 days. Scale bar: 100 μm.

Supplementary Fig. 15 (A) Immunofluorescence images of tumor slices stained by the hypoxia probe. The nuclei and hypoxia areas were stained by DAPI (blue) and anti-pimonidazole antibody (green), respectively. Scale bar: 200 μm . (B) Quantification of hypoxia area of tumor slices according to (A).

“The tumors were significantly suppressed in the group of (MSNs-ICG)@LA and an 808 nm laser due to the therapeutic effect of CDT/PDT, and the tumors in the group treated with (MSNs@CaO₂-ICG)@LA and an 808 nm laser irradiation were completely eliminated owing to the strongest ROS generation in H₂O₂/O₂ self-supplying CDT/PDT (Fig. 5G and Supplementary Fig. 15). The corresponding H&E and TUNEL staining of tumor slides also displayed the maximum tumor necrosis and apoptosis (Fig. 5H).”

(7) The biodistribution of (MSNs@CaO₂-ICG)@LA in blood circulation should be first measured.

Response: Thank you for your advice. We have measured the biodistribution of (MSNs@CaO₂-ICG)@LA in blood circulation added as Fig. 5A and revised the manuscript as follow:

Fig. 5 (A) Blood circulation and (B) biodistribution of (MSNs@CaO₂-ICG)@LA by measuring Mn concentrations over a span of 24 h after intravenous injection. (C) In vivo MRI of MCF-7 tumor-bearing mice with different treatments after 24 h (Laser: 808 nm, 0.64 W/cm², 10 min). (D) The body weight and (E) the relative cancer volume changes of MCF-7 tumor-bearing mice after various treatments in 14 days (**p < 0.01 and ***p < 0.001, two-tailed t test). (F) Representative photos of the mice with different treatments and corresponding tumor tissues collected from different groups at 14 days. (G) DCFH (green)- and DAPI (blue)-costained tumor tissues collected from different groups at 24 h post-injection. Scale bar: 200 μm. (H) H&E- or TUNEL-stained tumor slices collected from different groups at 14 days. Scale bar: 100 μm.

“The antitumor efficacy of (MSNs@CaO₂-ICG)@LA-mediated CDT/PDT synergistic therapy in vivo was further conducted in MCF-7 tumor-bearing mice. **Fig. 5A** exhibited the blood circulation curve of (MSNs@CaO₂-ICG)@LA in mice by measuring the concentrations of Mn²⁺ in the blood at different time points after tail vein injection, and the fairly long circulation time was favorable for effective tumor accumulation of (MSNs@CaO₂-ICG)@LA in tumor site due to the enhanced permeability and retention (EPR) effect which was observed in **Fig. 5B**.”

“In vivo blood circulation and biodistribution. MCF-7 cancer-bearing female mice were intravenously injected with (MSNs@CaO₂-ICG)@LA (10 mg/kg). At indicated time points (0.1, 0.5, 1, 2, 4, 6, 11, 24 h) after intravenous injection, 50 μL blood was extracted from each mouse. After intravenous injection for 24 h, the mice were sacrificed to measure the Mn amount in liver, spleen, kidney, heart, lung, tumor as well as blood samples by ICP-OES.”

(8) *In figure 3, the released Mn concentration and GSH/GSSG ratio should also be examined.*

Response: Thank you for your valuable advice. We have added the measurement of released Mn concentration as shown in Supplementary Table 1 and revised the manuscript. In addition, we tested the GSH/GSSG ratio in MCF-7 cells before and after incubation with MSNs (50 μg/mL) for 12 h using GSH and GSSG Assay Kit (Beyotime), which was 12.20 and 7.57 respectively. The decreased GSH/GSSG ratio indicated the GSH depletion property of MSNs. However, oxidants interfere with the determination of this kit because redox reactions are involved in the detection of this kit. Therefore, we cannot test the GSH/GSSG ratio in MCF-7 cells treated with (MSNs@CaO₂-ICG)@LA.

Supplementary Table 1. Measurement of cumulative Mn by ICP-MS in different system after reaction for

1 h.

Sample	MSNs		(MSNs@CaO ₂ -ICG)@LA	
Treatments	+ 0 mM GSH	+ 10 mM GSH	+ 10 mM GSH	+ 10 mM GSH + 808 nm laser
Released Mn (mM)	0.026	0.44	0.019	0.46
Released Mn (%)	5.20	87.98	3.74	92.27

The initial concentration of Mn in all samples was 0.5 mM.

“The released Mn²⁺ from MSNs could also be utilized as MRI contrast agent. As shown in Fig. 3E, the T₁ signal intensity of MSNs (group □) had negligible change and the released Mn²⁺ were at very low concentration, while MSNs treated with 10

mM GSH (group □) exhibited enhanced brightness derived from paramagnetic Mn²⁺ centers because MSN was reduced by GSH and the increasing free Mn ions (Supplementary Table 1) were easier to proceed chemical exchange with protons than isolated Mn centers in MSNs for enhanced T₁ signal³⁴. Thus, it was rational that (MSNs@CaO₂-ICG)@LA displayed enhanced brightness only when co-treated with GSH and NIR laser irradiation (group □). Remarkably, the longitudinal relaxivity coefficient (r₁) of group □ and corresponding released Mn concentration were larger than that of group □ (Fig. 3F). This was attributed to the enhanced release of Mn²⁺ from MSNs accelerated by photothermal effect of ICG (Supplementary Table 1).”

(9) In figure 4, images of hypoxic marker should be stained to show the relief of oxygen by the nanosystem.

Response: Thank you for your valuable advice. We have used ROS probe and hypoxic marker to co-stain the MCF-7 cells added as Fig. 4E and revised the manuscript as follow:

Fig. 4 (A) Cell viability of MCF-7, A549 and NHDF cells treated with different concentrations of (MSNs@CaO₂-ICG)@LA by CCK-8 assay. (B) The rate of MCF-7 cells uptaking (MSNs@CaO₂-ICG)@LA ([MSNs]= 25 µg/mL) and (C) corresponding mean fluorescence intensity by flow cytometry after different incubation time. (D) Fluorescence images showing cellular uptake of (MSNs@CaO₂-ICG)@LA ([MSNs]= 25 µg/mL) in MCF-7 cells after incubation for 4 h. Scale bar: 20 µm. (E) Fluorescence images showing ROS and hypoxia level in MCF-7 cells with different treatment under normoxia condition. ([MSNs]=25 µg/mL. Laser: 0.64 W/cm², 10 min). Scale bar: 50 µm. (F) Cell viability of MCF-7 cells with different treatments under hypoxia or normoxia condition ([MSNs]=25 µg/mL. Laser: 0.64 W/cm², 10 min) (**p < 0.01 and ***p < 0.001, two-tailed t test). (G) Fluorescence images of Calcein-AM- and propidium iodide (PI)-costained MCF-7 cells with different treatments under normoxia condition ([MSNs]=25 µg/mL. Laser: 0.64 W/cm², 10 min). Scale bar: 200 µm.

“.....As expected, the intracellular fluorescence imaging by ROS probe (DCFH-DA, green) and hypoxia probe demonstrated that the (MSNs@CaO₂-ICG)@LA-treated cells with irradiation displayed the strongest green fluorescence and slight enhanced red fluorescence compared with other groups both under normoxia or hypoxia conditions (Fig. 4E and Supplementary Fig. 12), indicating tremendous ROS generation enhanced by O₂ and H₂O₂ self-supply from CaO₂.

Intracellular hypoxia relief and ROS measurement. Adherent MCF-7 cells were incubated in normoxia (5% O₂, 21% CO₂, and 74% N₂) or hypoxia chamber (1% O₂, 5% CO₂, and 94% N₂) for 4 h to build normoxic or hypoxic environment. Then the cells were incubated with OPTI-MEM solution containing PBS (10 mM, pH = 7.4), MSNs, (MSNs-ICG)@LA or (MSNs@CaO₂-ICG)@LA for 4 h ([MSNs] = 25 µg/mL), washed with PBS (10 mM, pH = 7.4) and co-stained with 1 µL of DCFH-DA (10 mM) and 1 µL of hypoxia detection probe (1 mM) (Hypoxia Detection Kit, Enzo) for 15 min. After washed with PBS (10 mM, pH = 7.4) for three times, the cells were irradiated by a 808 nm laser (0.64 W/cm²) for 10 min and observed by CLSM.”

Reviewers' Comments:

Reviewer #1:

Remarks to the Author:

(1) The GSH concentrations in cells are affected by the glycolytic pathway. The authors should remove the effect of glycolytic pathway and consider whether the system has the GSH depletion.

(2) The CaO₂ nanoparticles is also pH-sensitive. Is the acidic environment in tumors affect the H₂O₂ generation? The H₂O₂ amount from (MSNs@CaO₂-ICG)@LA should be quantified in cells. Then the GSH depletion in cells should also be quantified.

(3) According to the previous report (Chem, 5, 1-12), the CaO₂ nanoparticles were proved to create an artificial calcium overloading stress in tumor cells, which is responsible for cell death. How do the authors distinguish the chemodynamic therapy results from calcium-overload-mediated tumor therapy?

(4) The authors have shown the thermal response of the (MSNs@CaO₂-ICG)@LA in vivo. The temperature was reported to reach to 46.8 °C, but the authors explained that it was not enough to harm the skin and tissue. Why? Many photothermal therapy associated-reports (such as Adv. Mater. 2016, 10155, Nat. Commun. 2017, 8, 14998, Small, 2018, 1702431, Angew. Chem. Int. Ed. 2018, 57, 246 and so on) can also reach to the similar temperature, which will definitely result in the skin burning and the observation of the black scars.

(5) H₂O₂ will generate oxygen and •OH under Mn²⁺ catalysis. So which one is more dominant in your system? Since the (MSNs@CaO₂-ICG)@LA is the final sample used for the therapy. In revised Figure S10B and C, the authors should use (MSNs@CaO₂-ICG)@LA for the testing. The singlet oxygen and heat caused by ICG under NIR laser irradiation will affect MB degradation, and these effects will exist in in vitro and vivo experiments. How the authors exclude this interruption?

Reviewer #2:

Remarks to the Author:

The authors have well addressed my comments.

Reviewer #3:

Remarks to the Author:

In this revised manuscript, the authors had well addressed all the questions raised by reviewers by either providing additional data figures or literatures. The MS is potentially accepted before one minor issue was resolved.

1. In figure 2 and supplementary figure 3, the scale bars for TEM images were inconsistent.

Dear Reviewers,

Thank you very much for the comments on the present work. Those comments are valuable and very helpful for revising and improving our paper, as well as the important guiding significance to our researches. In the following we make a point-to-point response to all the comments.

Detailed list of changes made and responses:

Response to Reviewer #1

(1) *The GSH concentrations in cells are affected by the glycolytic pathway. The authors should remove the effect of glycolytic pathway and consider whether the system has the GSH depletion.*

Response: Thank you for your comments. Since GSH participates as a cellular protection agent by controlling a variety of cellular processes including cell differentiation, metabolism, antioxidant defense and carcinogenicity balance, GSH concentrations in cells are affected by many factors rather than only glycolytic pathway. In our paper, we used various methods including exocellular and intracellular experiments to prove the GSH depletion property: 1. Observation of the degradation of MSNs under different concentration of GSH (Supplementary Fig. 3). 2. Measurement of the concentration of released Mn ions from MSNs (Supplementary Table 1). 3. Measurement of the GSH/GSSG ratio in MCF-7 cells before and after incubation with MSNs for 12 h using GSH and GSSG Assay Kit (12.20 and 7.57 respectively). These results suggest the redox reaction between MSNs and GSH and indicate the GSH depletion property of MSNs, which also have been confirmed in previous reports (J. Am. Chem. Soc. 2016, 138, 9881-9894; Angew. Chem. Int. Ed. 2018, 57, 4902-4906; ACS Nano 2019, 13, 4267-4277; Acta Biomater. 2016, 30, 378-387).

(2) *The CaO₂ nanoparticles is also pH-sensitive. Is the acidic environment in tumors affect the H₂O₂ generation? The H₂O₂ amount from (MSNs@CaO₂-ICG)@LA should be quantified in cells. Then the GSH depletion in cells should also be quantified.*

Response: Thank you for your comments. It is right that the CaO_2 nanoparticles is pH-sensitive and decreasing pH can accelerate H_2O_2 release rate (Chem. 2019, 5, 2171-2182; Chemical Engineering Journal 2016, 303, 450-457). We have measured the concentration of released H_2O_2 from CaO_2 under different pH condition added as Supplementary Fig. 6, and confirmed the acidic environment is good for the H_2O_2 generation. Related discussions have been added in the revised manuscript (Page 8, paragraph 1). As for the quantification of H_2O_2 amount from $(\text{MSNs@CaO}_2\text{-ICG})\text{@LA}$ in cells, we are afraid that we cannot perform the experiments in this special time due to the outbreak of COVID-19 in China.

Supplementary Fig. 6 H_2O_2 cumulative release from CaO_2 under different pH conditions. ($[\text{CaO}_2]=10$ $\mu\text{g/mL}$).

In addition, we tested the GSH/GSSG ratio in MCF-7 cells before and after incubation with MSNs (50 $\mu\text{g/mL}$) for 12 h using GSH and GSSG Assay Kit (Beyotime), which was 12.20 and 7.57 respectively. The decreased GSH/GSSG ratio indicated the GSH depletion property of MSNs. However, oxidants will interfere with the determination of this kit because redox reactions are involved in the detection of this kit. Therefore, we cannot test the GSH/GSSG ratio in MCF-7 cells treated with $(\text{MSNs@CaO}_2\text{-ICG})\text{@LA}$.

(3) According to the previous report (Chem, 5, 1-12), the CaO_2 nanoparticles were proved to create an artificial calcium overloading stress in tumor cells, which is

responsible for cell death. How do the authors distinguish the chemodynamic therapy results from calcium-overload-mediated tumor therapy?

Response: Thank you for your comments. The report you listed is nonexistent. According to your description, the reports may be Chem. 2019, 5, 2171-2182 (Calcium-Overload-Mediated Tumor Therapy by Calcium Peroxide Nanoparticles). It is right that CaO₂ NPs will slowly decompose into free Ca²⁺ and H₂O₂, and the oxidative stress resulting from excessive cellular H₂O₂ will lead to intracellular Ca²⁺ overload and subsequently cell death. In this report, the inhibition effect of CaO₂ NPs on cancer cells was dependent on the concentration of Ca²⁺ (Figure 3A, Chem. 2019, 5, 2171-2182), and the significant cytotoxicity was observed when the concentration of Ca²⁺ was over 40 µg/mL. However, in our paper, the concentration of Ca²⁺ in (MSNs@CaO₂-ICG)@LA used to study the inhibition effect on cancer cells was about 14 µg/mL, which was much lower and would not have obvious impact on the cancer cells according to the Figure 3A in the report. Therefore, the good tumor inhibition effect in our paper attribute to the hypoxia-relieved ICG-mediated PDT and H₂O₂-supplying MSN-based CDT, while the calcium overloading stress induced by CaO₂ NPs can be negligible and is not the focus of our paper.

(4) The authors have shown the thermal response of the (MSNs@CaO₂-ICG)@LA in vivo. The temperature was reported to reach to 46.8 °C, but the authors explained that it was not enough to harm the skin and tissue. Why? Many photothermal therapy associated-reports (such as Adv. Mater. 2016, 10155, Nat. Commun. 2017, 8, 14998, Small, 2018, 1702431, Angew. Chem. Int. Ed. 2018, 57, 246 and so on) can also reach to the similar temperature, which will definitely result in the skin burning and the observation of the black scars.

Response: Thank you for your comments. It has been reported that within defined temperature ranges tissue thermal damage is approximately linearly dependent upon exposure time and exponentially dependent upon the temperature elevation (Int. J. Hyperthermia 2005, 21, 745-53):

Plot of temperature-time thresholds of acute thermal damage measured by Henriques and Moritz in pig and human skin in vivo.

Therefore, only when the temperature is high enough and the duration of thermal exposure last a certain amount of time will critical cellular proteins, tissue structural components and the vasculature irreversibly damage and coagulate, leading to immediate tissue destruction (Int. J. Hyperthermia 2005, 21, 745-53).

In our paper, the tumor temperature of (MSNs@CaO₂-ICG)@LA-treated group was 43.5 °C at 5 min and 46.8 °C at 10 min (Supplementary Fig. 13). The temperature increasing rate was relatively slow, the final temperature was not high, and the generated heat was mainly absorbed to melt the phase-change material LA. Thus the heat was not enough to harm the skin and tissue.

(5) *H₂O₂ will generate oxygen and •OH under Mn²⁺ catalysis. So which one is more dominant in your system? Since the (MSNs@CaO₂-ICG)@LA is the final sample used for the therapy. In revised Figure S10B and C, the authors should use (MSNs@CaO₂-ICG)@LA for the testing. The singlet oxygen and heat caused by ICG under NIR laser irradiation will affect MB degradation, and these effects will exist in in vitro and in vivo experiments. How the authors exclude this interruption?*

Response: Thank you for your comments. Generally, in the presence of excess H₂O₂ and absence of other reagents such as organic substrates, H₂O₂ will produce a lot of oxygen under Mn²⁺ catalysis. However, in Fenton reaction, H₂O₂ will dominantly generate •OH to oxidize organic substrates and accompany the production of a small

production of O₂ under Mn²⁺ catalysis (Chem. Eur. J. 2003, 9, 3436-3444). In the Figure S10B and C, we aimed to use MB degradation experiments to preliminarily study the GSH depletion-enhanced and H₂O₂ concentration-dependent chemodynamic efficacy of MSNs respectively. As for that testing of (MSNs@CaO₂-ICG)@LA, we have used EPR testing (Fig. 3D) to further clarify the GSH depletion-enhanced and H₂O₂ concentration-dependent •OH generation considering the MB degradation may be interrupted by the ¹O₂ and heat caused by ICG under NIR irradiation. Lastly, the ¹O₂ and heat caused by ICG under NIR laser irradiation will not affect the in vivo experiments, because the ¹O₂ generated by ICG under NIR irradiation is part of ROS for tumor treatment, and the heat is negligible to affect the therapeutic results that we have explained in question (4).

Response to Reviewer #2

Comments: The authors have well addressed my comments.

Response: Thank you very much for your recommendation of our work.

Response to Reviewer #3

Comments: In this revised manuscript, the authors had well addressed all the questions raised by reviewers by either providing additional data figures or literatures. The MS is potentially accepted before one minor issue was resolved.

Response: Thank you very much for your recommendation of our work.

(1) In figure 2 and supplementary figure 3, the scale bars for TEM images were inconsistent.

Response: Thank you for your kind remind. We have carefully checked and revised the Supplementary Fig. 3 as follows:

Supplementary Fig. 3 The effect of GSH (0, 1, 10 mM) on biodegradation of MSNs (100 $\mu\text{g}/\text{mL}$) in PBS (10 mM, pH 7.4) for various periods of time. Scale bar: 100 nm.

Thank you once again for the reviewers' instructive comments on our work. If any of the responses are misunderstood, please inform us and we will reconsider them. All the corrections mentioned in this response are also addressed and highlighted in red in the revised manuscript.

Sincerely yours

Haifeng Dong and co-authors

Reviewers' Comments:

Reviewer #1:

Remarks to the Author:

I think that this manuscript is suitable for publication without further revision.